# Allosteric inhibition of trypanosomatid pyruvate kinases by a camelid single-domain antibody

Joar Esteban Pinto Torres[1], Mathieu Claes[2], Rik Hendrickx[2], Meng Yuan[3†], Natalia Smiejkowska[4], Pieter Van Wielendaele[4], Aysima Hacisuleyman[5], Hans De Winter[6], Serge Muyldermans[1], Paul AM Michels[3], Malcolm D Walkinshaw[3], Wim Versées[7,8], Guy Caljon[2], Stefan Magez[1,9,10†], Yann G-J Sterckx[2*†]

[1]Laboratory for Cellular and Molecular Immunology (CMIM), Vrije Universiteit Brussel (VUB), Brussel, Belgium; [2]Laboratory of Microbiology, Parasitology and Hygiene (LMPH) and the Infla-Med Centre of Excellence, University of Antwerp, Wilrijk, Belgium; [3]School of Biological Sciences, The University of Edinburgh, Edinburgh, United Kingdom; [4]Laboratory of Medical Biochemistry (LMB) and the Infla-Med Centre of Excellence, University of Antwerp, Wilrijk, Belgium; [5]Department of Computational Biology, University of Lausanne, Lausanne, Switzerland; [6]Laboratory of Medicinal Chemistry, University of Antwerp, Wilrijk, Belgium; [7]VIB-VUB Center for Structural Biology, VIB, Brussels, Belgium; [8]Structural Biology Brussels, Vrije Universiteit Brussel, Brussels, Belgium; [9]Center for Biomedical Research, Ghent University Global Campus, Incheon, Republic of Korea; [10]Department for Biochemistry and Microbiology, Ghent University, Ghent, Belgium

*For correspondence:
yann.sterckx@uantwerpen.be

†These authors contributed equally to this work

## eLife Assessment

This work presents **valuable** data demonstrating that a camelid single-domain antibody can selectively inhibit a key glycolytic enzyme in trypanosomes via an allosteric mechanism. The claim that this information can be exploited for the design of novel chemotherapeutics is **solid** but limited by the modest effects on parasite growth, as well as the lack of evidence for cellular target engagement in vivo.

**Abstract** African trypanosomes are the causative agents of neglected tropical diseases affecting both humans and livestock. Disease control is highly challenging due to an increasing number of drug treatment failures. African trypanosomes are extracellular, blood-borne parasites that mainly rely on glycolysis for their energy metabolism within the mammalian host. Trypanosomal glycolytic enzymes are therefore of interest for the development of trypanocidal drugs. Here, we report the serendipitous discovery of a camelid single-domain antibody (sdAb aka Nanobody) that selectively inhibits the enzymatic activity of trypanosomatid (but not host) pyruvate kinases through an allosteric mechanism. By combining enzyme kinetics, biophysics, structural biology, and transgenic parasite survival assays, we provide a proof-of-principle that the sdAb-mediated enzyme inhibition negatively impacts parasite fitness and growth.

## Introduction

Neglected tropical diseases (NTDs) comprise a wide variety of communicable diseases that are prevalent in (sub)tropical regions and affect more than 1 billion people worldwide. It is becoming increasingly clear that NTDs constitute a major health threat in both developing and developed countries, with those living in poverty being especially vulnerable (*Hunter, 2014*; *Picado et al., 2019*). NTDs are typically characterized by a low mortality and high morbidity, which results in a severe impact on the quality of life and economic productivity of those affected. In recent times, NTD control has become more complicated by globalization, human migration, climate change and the altered distribution of disease-transmitting vectors (including mosquitoes, flies, and ticks). Consequently, even currently unaffected areas (including the Western world) are confronted with the (re-)emergence of NTDs. The WHO has listed 20 NTDs that should be tackled in the interest of global health and well-being (https://www.who.int/neglected_diseases/diseases/en). Three of these are caused by trypanosomatids, a group of flagellated, single-celled eukaryotic organisms comprising parasites of the *Trypanosoma* and *Leishmania* genera. *Trypanosoma* spp. encompass African trypanosomes, which are extracellular parasitic protists that cause human and animal trypanosomoses (HAT and AAT, respectively).

To be effective, the battle against trypanosomes requires a concerted approach including vaccination, drug treatment, and vector control. However, the development of an effective vaccine against these parasites is thwarted by sophisticated immune-evasion strategies (*Pays et al., 2023*), while vector control may be hampered due to resistance of the insect vector to insecticides (*Field et al., 2017*). As a result, chemotherapy is an essential pillar for clinical management, control and/or elimination. Unfortunately, a number of reports describe treatment failure or parasite resistance to the currently available drugs (*De Rycker et al., 2018*). Hence, there is a need for alternative compounds, preferably with novel modes of action and/or designed based on mechanistic insights of the target's structure-function relationship (*Field et al., 2017*; *De Rycker et al., 2018*). This need is especially pressing for AAT, which strongly impedes sustainable livestock rearing in Sub-Saharan Africa. AAT results in drastic reductions of draft power, meat, and milk production by the infected animals (small and large ruminants), and its control relies mainly on vector control and chemotherapy, with only few drugs currently available. The lack of routine field diagnosis has resulted in the misuse of trypanocidal drugs, thereby accelerating the rise of parasite resistance and further exacerbating the problem (*Richards et al., 2021*). As such, AAT-inflicted annual losses are estimated at around $ 5 billion (and the necessity to invest another $ 30 million each year to control AAT through chemotherapy), thereby having a devastating impact on the socio-economic development of Sub-Saharan Africa (*Fetene et al., 2021*). In contrast, HAT is perceived as a minor threat as it has reached a post-elimination phase as a public health problem with less than 1000 yearly documented cases (*Franco et al., 2022*). In addition, new and effective drugs for HAT treatment have recently become available (*De Rycker et al., 2023*). HAT control currently relies on case detection and treatment, and vector control (*Büscher et al., 2017*).

African trypanosomes are extracellular parasites that have a bipartite life cycle involving insect vectors and mammals as hosts (*Radwanska et al., 2018*). Most HAT (*T. brucei gambiense* and *T. b. rhodesiense*) and AAT (*T. b. brucei* and *T. congolense*) causing trypanosomes are uniquely vectored by tsetse flies (*Glossina* spp.) and are confined to Sub-Saharan Africa. *T. b. evansi* and *T. vivax* (both causative agents of AAT) have expanded beyond the tsetse belt due to their ability to be mechanically transmitted by a variety of biting flies (*Glossina*, *Stomoxys*, and *Tabanus* spp.). Finally, *T. b. equiperdum* infects equids and represents an exception as it is transmitted directly from animal to animal through sexual contact. For all African trypanosomes, the predominant parasite form within the mammalian host is called the bloodstream form (BSF) which, as the name suggests, resides mainly inside the host bloodstream. The BSF also colonizes sites such as the lymphatics, the skin, brain, testes, adipose tissue, and lungs (*Trindade et al., 2016*; *Caljon et al., 2016*; *Capewell et al., 2016*; *Krüger et al., 2018*; *Mabille et al., 2022*). Given its niche, the BSF has steady access to high blood glucose concentrations (~5 mM) and has evolved to exclusively rely on glycolysis to power its metabolism. For this reason, trypanosomal glycolytic enzymes (of which most are localized in organelles called glycosomes *Szöör et al., 2014*; *Haanstra et al., 2016*; *Figure 1A*) have received much interest and attention as targets for the development of trypanocidal compounds (*Bakker et al., 2000*; *Verlinde et al., 2001*; *Haanstra and Bakker, 2015*). Indeed, informed by computational models of trypanosomal glycolysis, RNAi experiments have shown that a reduction in glycolytic flux induces growth impairment and

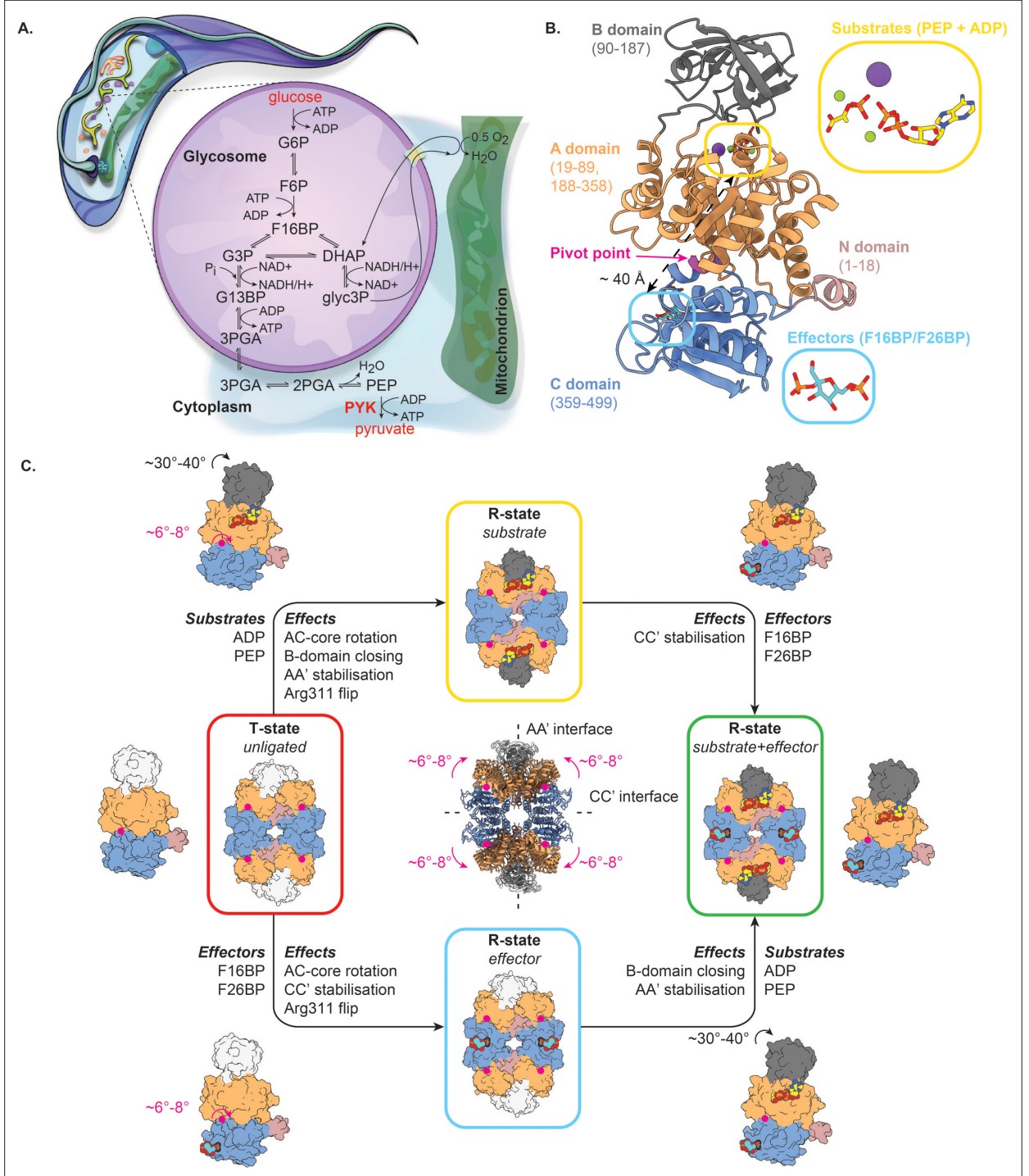

**Figure 1.** Structure-function relationship of trypanosomatid PYKs. (**A**) Schematic representation of a trypanosome, with a focus on their glycosome biochemistry. PYK catalyzes the last reaction of the trypanosomal glycolysis and is located outside of the glycosomes. (**B**) The PYK monomer with the different domains color-coded and the domain boundaries shown. The pivot point for the AC core rotation (residues 430–433) is indicated by a magenta arrow. The substrate and effector binding sites are highlighted by yellow and cyan boxes, respectively. (**C**) Schematic representation of the 'rock and lock' model. The different PYK domains are color-coded as in panel (**B**). In the absence of substrates (PEP and ADP) and effectors (F26BP or F16BP), trypanosomatid PYKs reside in a T-state (red box). The binding of substrates causes the enzyme to 'rock' (R-state boxed in yellow). This consists of several structural rearrangements across the entire PYK tetramer that involve (i) AC-core rotation of 6°–8° (with residues 430–434 as a pivot point), (ii) closing of the lid domain (rotation of 30°–40°), (iii) stabilisation of the AA' dimer interfaces, and (iv) flipping of the Arg311 side chain as part of remodeling the catalytic pocket for substrate accommodation. The binding of effectors to PYK's C domain generates a 'lock' in addition to the 'rock'.

*Figure 1 continued on next page*

*Figure 1 continued*

This prompts the enzyme to adopt a conformation primed for efficient catalysis (R-state boxed in blue); this involves (i) the 6°–8° AC-core rotation, (ii) stabilisation of the CC' dimer interfaces, and (iii) the Arg311 flip. The presence of substrates and effectors "rock and lock" the enzyme in the R-state (green box). The ribbon representations in the inset are the tetramer structures of T and R state PYK, superposed on the four pivot points. The AA' and CC' dimer interfaces are indicated by dashed lines. All structures and schematics were based on the crystal structures of apo and holo *Tco*PYK (this work and *Pinto Torres et al., 2020*).

eventually leads to parasite death (*Albert et al., 2005*; *Haanstra et al., 2011*). This principle was recently successfully exploited by McNae, Kinkead, Malik and co-workers, who developed a novel and selective small-molecule inhibitor of the trypanosomal glycolytic enzyme phosphofructokinase (PFK; *McNae et al., 2021*). The compound impairs PFK activity via an allosteric mechanism and was validated to lead to parasite death in vitro as well as in an in vivo mouse model. Pyruvate kinase (PYK) represents another attractive glycolytic target. This non-glycosomal enzyme catalyses the last step of the glycolysis (the irreversible conversion of phosphoenolpyruvate (PEP) to pyruvate; *Figure 1A*). The importance of this reaction is twofold: (i) the generation of ATP through the transfer of a phosphoryl group from PEP to ADP and (ii) the formation of pyruvate, a crucial metabolite of the central metabolism. Like most PYKs, trypanosomatid PYKs are homotetramers. The PYK monomer is a ~55 kDa protein organized into four domains termed 'N', 'A', 'B', and 'C' (*Figure 1B*). The A domain constitutes the largest part of the PYK monomer and is characterized by an $(\alpha/\beta)_8$-TIM barrel fold that contains the active site. Together with the N-terminal domain, it is also involved in the formation of the PYK tetramer AA' dimer interfaces. The B domain is known as the flexible 'lid' domain that shields the active site during enzyme-mediated phosphotransfer. Finally, the C domain harbors the binding pocket for allosteric effectors and stabilizes the PYK tetramer by formation of CC' dimer interfaces. Because of their role in ATP production and distribution of fluxes into different metabolic branches, the activity of trypanosomatid PYKs is tightly regulated through an allosteric mechanism known as the 'rock and lock' model (*Morgan et al., 2010*; *Morgan et al., 2014*; *Pinto Torres et al., 2020*). In this model (which is detailed in *Figure 1C*), the binding of substrates and/or effectors (and analogs thereof) to the active and effector sites, respectively, trigger a conformational change from the enzymatically inactive T state to the catalytically active R state. Known effector molecules for trypanosomatid PYKs are fructose 2,6-bisphosphate (F26BP), fructose 1,6-bisphosphate (F16BP) and sulfate ($SO_4^{2-}$), with F26BP being the most potent one (*van Schaftingen et al., 1985*; *Callens and Opperdoes, 1992*; *Ernest et al., 1994*; *Tulloch et al., 2008*). Interestingly, trypanosomatid PYKs seem to be largely unresponsive to the allosteric regulation of enzyme activity by free amino acids (*Callens et al., 1991*), which contrasts with human PYKs (*Chaneton et al., 2012*; *Yuan et al., 2018*). Known trypanosomatid PYK inhibitors impair enzymatic activity through occupation of the PYK active site (*Morgan et al., 2011*). Here, we describe an allosteric mechanism for the inhibition of trypanosomatid PYKs, which was identified through a camelid single-domain antibody (sdAb aka nanobody) raised against *T. congolense* PYK (*Tco*PYK). By using a combination of enzyme kinetics assays, circular dichroism (CD)

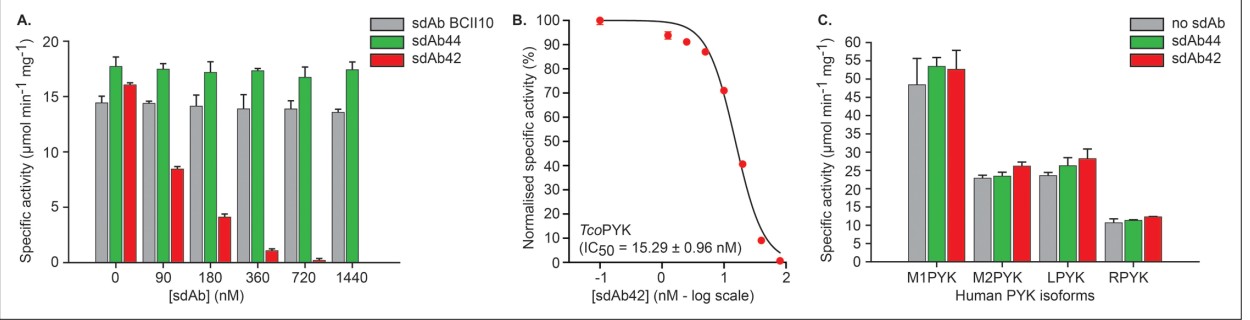

**Figure 2.** sdAb42 selectively inhibits TcoPYK. (**A**) Effect of the addition of various concentrations of sdAb42 (red bars), sdAb44 (green bars), or sdAb BCII10 (grey bars) on the activity of *Tco*PYK prior to addition of substrates and effectors. The results demonstrate that only sdAb42 abrogates *Tco*PYK activity. (**B**) IC$_{50}$ determination for sdAb42 tested against *Tco*PYK, for which three independent inhibition assay replicates were performed. (**C**) Effect of the addition of sdAb42 (red bars) or sdAb44 (green bars) at a 1000-fold molar excess on the activities of the various human PYK isoforms (M1PYK, human skeletal muscle isoform 1; M2PYK, human skeletal muscle isoform 2; LPYK, human liver; RPYK, human red blood cell). No impact of either sdAbs on enzyme activity could be observed.

spectroscopy and macromolecular X-ray crystallography (MX), we demonstrate that sdAb42 (one of the sdAbs raised against *Tco*PYK) is a potent inhibitor that impairs enzyme function by selectively binding and stabilizing the enzyme's inactive T state. The sdAb42 epitope is located far away from the active and effector sites, and perturbation analysis (*Wang et al., 2020*) further reveals the epitope contains residues that are characterized by (i) a high allosteric coupling intensity to the active site and (ii) critical components of the allosteric communication pathway between the *Tco*PYK effector and active sites. In addition, we show that the inhibitory mechanism of sdAb42 applies to trypanosomatid PYKs in general, as its epitope is highly conserved among *Trypanosoma* and *Leishmania* parasites. Finally, we provide evidence that the production of sdAb42 as an 'intrabody' (intracellularly produced sdAb) has a modest effect on parasite growth in transgenic *T. brucei* lines.

## Results

### sdAb42 is a potent and selective *Tco*PYK inhibitor

We previously identified *Tco*PYK as a biomarker for the detection of active *T. congolense* infections by immunoassays, using a pair of camelid sdAbs (sdAb42 and sdAb44; *Pinto Torres et al., 2018*). Because the target of these two sdAbs is a trypanosomal glycolytic enzyme, we sought to investigate whether they had the potential to inhibit *Tco*PYK enzymatic activity. While complexation of *Tco*PYK with sdAb44 and a negative control sdAb (BCII10) have no impact, the addition of increasing sdAb42 concentrations severely reduces (and even completely abolishes) enzymatic activity with an $IC_{50}$ value of around 15 nM (*Figure 2*, panels A and B). Moreover, the inhibition displayed by sdAb42 is selective for *Tco*PYK as no effect on the enzymatic activity of human PYK could be observed (even at a 1000-fold sdAb excess, *Figure 2C*). In conclusion, sdAb42 appears to be a potent, selective *Tco*PYK inhibitor.

### sdAb42 impairs *Tco*PYK activity by selectively binding and stabilising the enzyme's inactive T state

To gain insights into the structural basis for the selective inhibition of *Tco*PYK by sdAb42, the high-resolution structure of the sdAb42:*Tco*PYK complex was determined through MX. A general overview of the crystal structure reveals that four sdAb42 molecules are bound to the *Tco*PYK tetramer (*Figure 3A*), which is in accordance with the stoichiometry in solution, previously determined via analytical gel filtration (*Pinto Torres et al., 2018*). Interestingly, the epitope of a single sdAb42 spans a region across two *Tco*PYK subunits linked together along the AA' dimer interface (*Figure 3B*). While CDR3 contacts residues from both subunits, CDR1 and CDR2 each exclusively interact with amino acids from the A' and A subunit domains, respectively (*Table 1*). Interactions are also provided by the flexible *Tco*PYK B domain, although these could not be observed in all copies of the asymmetric unit as not all B domains could be built due to lack of electron density. A comparison of the conformation of sdAb42-bound *Tco*PYK to the enzyme's R state structure and an inspection of the tell-tale features of T and R state trypanosomatid PYK conformations (*Figure 1C*) show that the enzyme resides in its inactive T state when interacting with sdAb42 (the signature Arg311 flip is displayed in *Figure 3C*).

Based on the above-mentioned sdAb42:*Tco*PYK crystal structure, we hypothesised that sdAb42 inhibits *Tco*PYK activity by selectively binding and stabilizing the enzyme's inactive T state. A first finding that supports this idea is that the sdAb42 epitope is significantly distorted when *Tco*PYK transitions from the T to the R state (Cα RMSD of 3.70 Å, *Figure 3D*). Second, we were also able to crystallise the sdAb42:*Tco*PYK complex in the presence of sulfate, which acts as a phosphate mimic that can bind both the active and effector sites of trypanosomatid PYKs. As a result, sulfate binding has the potential to initiate the 'rock and lock', thereby ushering the transition from the inactive T to the active R state (*Tulloch et al., 2008*). A detailed inspection of the electron density revealed the presence of sulfate molecules in the *Tco*PYK effector site at the positions usually occupied by the phosphoryl groups of the cognate effector molecules fructose 1,6-bisphosphate and fructose 2,6-bisphosphate (*Figure 3—figure supplement 1*). Despite the presence of these sulfates, *Tco*PYK clearly remains in a T state conformation when bound by sdAb42. Third, thermal unfolding followed by CD spectroscopy shows that sdAb42 significantly stabilises apo *Tco*PYK (*Figure 3E*). In accordance with our previous findings (*Pinto Torres et al., 2020*), apo *Tco*PYK displays an apparent melting temperature ($T_{m,app}$) of ~46 °C. The binding of sdAb42 leads to a remarkable increase in the enzyme's thermal stability ($\Delta T_{m,app}$

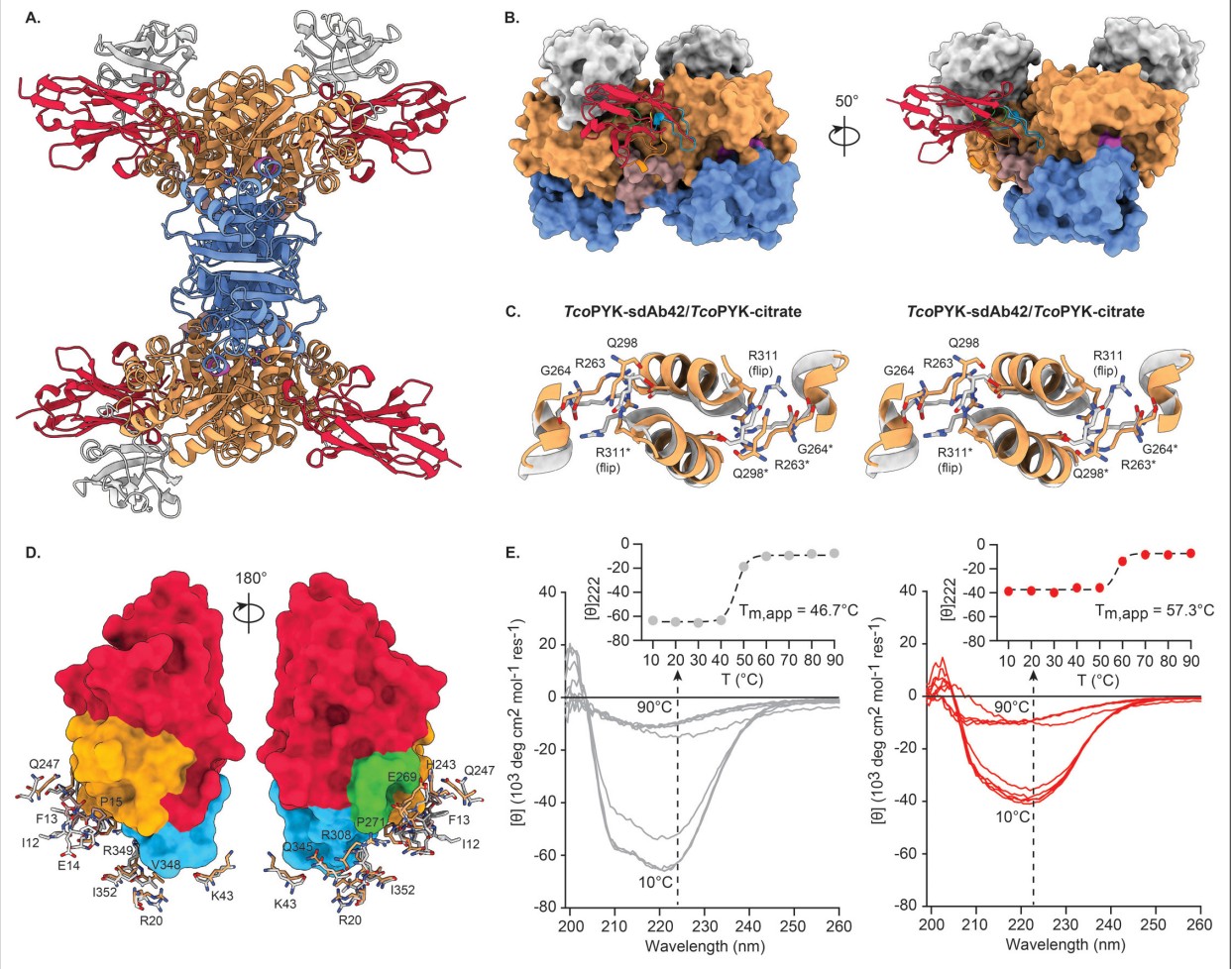

**Figure 3.** sdAb42 binds and stabilizes the *Tco*PYK T state. (**A**) Cartoon representation of the sdAb42-*Tco*PYK complex observed in the crystal, in which one TcoPYK tetramer is bound by four copies of sdAb42. The TcoPYK domains are color-coded as in *Figure 1* and sdAb42 is depicted in red. (**B**) Close-up of the interaction between a single sdAb42 copy (cartoon representation) and AA' dimer interface *Tco*PYK subunits (surface representation). sdAb42 and TcoPYK are color-coded as in panel (**A**). The sdAb42 CDR1, CDR2, and CDR3 are colored in blue, green, and orange, respectively. (**C**) Stereo view of the signature interactions made by Arg311 at the AA' interface for *Tco*PYK in its T (TcoPYK-sdAb42, colored as in panels (**A**) and (**B**)) and R state (*Tco*PYK-citrate, colored in light grey; PDB ID 6SU1 *Pinto Torres et al., 2020*). Residues Arg263, Gly264, Gln298, Arg311, and Asp316 are shown in stick representation. The residues originating from the A' domain are indicated by an asterisk '*'. (**D**) Detailed view of the sdAb42 epitope in T and R state *Tco*PYK. sdAb42 is shown in surface representation and color-coded as in panel (**A**). The residues constituting the sdAb42 epitope are shown in stick representation and colored in light grey (R state *Tco*PYK) or color-coded as in panels (**A**) and (**B**) (T state *Tco*PYK). A residue-by-residue comparison reveals that the epitope is significantly distorted in R state *Tco*PYK. (**E**) CD spectra of apo *Tco*PYK (left panel, grey traces) and the sdAb42:*Tco*PYK complex (right panel, red traces) collected at different temperatures. The black dotted arrow represents the effect of the increasing temperature on the mean residue ellipticity measured at 222 nm, plotted in the inset (filled circles and dashed line represent the experimental data points and fit, respectively).

The online version of this article includes the following figure supplement(s) for figure 3:

**Figure supplement 1.** Comparison of the effector binding sites of different *Tco*PYK structures.

= 10.3°C). When taken together, the data strongly indicate that sdAb42 impairs *Tco*PYK activity by selectively binding and stabilising the enzyme's inactive T state.

## The sdAb42 epitope contains residues that are critical for the allosteric communication between the enzyme's effector and active sites

To better understand the mechanistic basis of *Tco*PYK inhibition by sdAb42, we performed in silico perturbation analyses, which allows (i) prediction of allosteric correlations between residue pairs, (ii) prediction of residues with a high allosteric coupling intensity (ACI) to the active site (and thus allosteric

**Table 1.** List of interactions between sdAb42 and *Tco*PYK.

The # symbol indicates the number of times the interaction was observed over the total of six sdAb42:*Tco*PYK complexes present in the asymmetric unit. The average distances are only given for hydrogen bonds or electrostatic interactions.

| sdAb42 | | | *Tco*PYK | | | | |
| --- | --- | --- | --- | --- | --- | --- | --- |
| **Residue** | **Group** | **FR/CDR** | **Residue** | **Group** | **Domain** | **Interaction (distance in Å)** | **# (out of 6)** |
| Ser27 | side chain (OG) | CDR1 | Gln345 | side chain (NE2) | A′ | H bond (3.08 ± 0.30) | 3 |
| Phe29 | side chain | CDR1 | Lys43 | side chain | A′ | Van der Waals | 6 |
| | | | Ser44 | backbone | A′ | Van der Waals | 4 |
| | | | Gln345 | side chain | A′ | Van der Waals | 2 |
| Ser30 | backbone (CO) | CDR1 | Arg20 | side chain (NH1) | A′ | H-bond (3.49) | 1 |
| Ser31 | backbone | CDR1 | Val348 | side chain | A′ | Van der Waals | 5 |
| | | | Arg349 | side chain | A′ | Van der Waals | 3 |
| | side chain | | Ile352 | side chain | A′ | Van der Waals | 5 |
| Gly32 | backbone | CDR1 | Gln345 | side chain | A′ | Van der Waals | 4 |
| | | | Val348 | side chain | A′ | Van der Waals | 5 |
| | | | Arg349 | side chain | A′ | Van der Waals | 6 |
| | backbone (CO) | | Arg349 | side chain (NH1) | A′ | H-bond (3.13 ± 0.36) | 3 |
| Thr34 | side chain | CDR1 | Gln345 | side chain | A′ | Van der Waals | 4 |
| | | | Arg349 | side chain | A′ | Van der Waals | 1 |
| Thr37 | side chain | CDR1 | Arg349 | side chain | A′ | Van der Waals | 6 |
| Trp59 | side chain (NE1) | CDR2 | Glu269 | backbone (CO) | A | H-bond (3.04 ± 0.23) | 5 |
| | side chain | | Ile270 | backbone | A | Van der Waals | 1 |
| | | | Pro271 | side chain | A | hydrophobic effect | 6 |
| Asn60 | backbone (CO) | CDR2 | Tyr143 | side chain (OH) | A | H-bond (3.63 ± 0.03) | 2 |
| | side chain (ND2) | | Val268 | backbone (CO) | A | H-bond (3.28 ± 0.26) | 6 |
| Gly61 | backbone | CDR2 | Tyr143 | side chain | A | Van der Waals | 1 |
| | | | Pro181 | side chain | A | Van der Waals | 3 |
| Gly62 | backbone | CDR2 | Tyr143 | side chain | A | Van der Waals | 1 |
| | | | Pro181 | side chain | A | Van der Waals | 2 |
| Ile63 | side chain | CDR2 | His243 | side chain | A | hydrophobic effect | 6 |
| | | | Glu269 | side chain | A | hydrophobic effect | 6 |
| Thr64 | side chain (OG1) | FR | Pro181 | backbone (CO) | A | H-bond (3.33 ± 0.41) | 3 |
| | side chain | | Gly182 | backbone | A | Van der Waals | 1 |
| | side chain (OG1) | | Cys183 | side chain (SG) | A | H-bond (3.32 ± 0.21) | 3 |
| Arg105 | side chain (NH1) | CDR3 | Phe13 | backbone (CO) | N′ | H-bond (3.25 ± 0.48) | 6 |
| | side chain | | Pro15 | side chain | N′ | hydrophobic effect | 5 |
| Asp106 | side chain | CDR3 | Pro15 | side chain | N′ | Van der Waals | 6 |
| Trp108 | side chain | CDR3 | Ile12 | side chain | N′ | hydrophobic effect | 6 |
| | side chain (NE1) | | Ile12 | backbone (CO) | N′ | H-bond (3.01 ± 0.15) | 6 |
| | side chain | | Phe13 | side chain | N′ | hydrophobic effect | 4 |
| | | | His243 | side chain | A | hydrophobic effect | 4 |
| | | | Ile270 | side chain | A | hydrophobic effect | 6 |
| | | | Pro271 | side chain | A | hydrophobic effect | 6 |

*Table 1 continued on next page*

*Table 1 continued*

| sdAb42 | | | TcoPYK | | | | |
| Residue | Group | FR/CDR | Residue | Group | Domain | Interaction (distance in Å) | # (out of 6) |
| --- | --- | --- | --- | --- | --- | --- | --- |
| | | | Lys274 | side chain | A | hydrophobic effect | 5 |
| Tyr109 | side chain | CDR3 | Phe13 | side chain | N' | hydrophobic effect | 4 |
| | | | His243 | side chain | A | hydrophobic effect | 4 |
| | side chain (OH) | | Gln247 | side chain (NE2) | A | H-bond (4.04 ± 0.35) | 3 |

sites or 'allosteric hotspots'), (iii) identification of allosteric communication pathways between an enzyme's active site and known effector sites, and (iv) identification of critical residues within these pathways (**Wang et al., 2020**).

The ACI predictions performed on the *Tco*PYK T and R state tetramer structures reveal that the sdAb42 epitope largely coincides with an 'allosteric hotspot' located on the enzyme's surface (**Figure 4A**). Especially the *Tco*PYK residues contacted by sdAb42's CDR1 (Arg20, Ser44, Val348, Ile352) display high ACI values (>0.75) implying that a perturbation of these residues will have a high probability of propagating to the active site and, hence, affecting enzyme activity. Interestingly, the ACI values of these residues are higher in the T state compared to the R state tetramer, which implies that the allosteric coupling between the sdAb42-recognized 'allosteric hotspot' and the enzyme's active site appears to be stronger in the T state conformation. This finding is further supported through the 'Allosteric Pocket Prediction (APOP)' method, which is employed for the identification of allosteric binding pockets (**Kumar et al., 2023**). The APOP analysis identifies sites overlapping with the sdAb42 epitope as highly ranking allosteric ligand binding pockets (**Figure 4—figure supplement 1**), thereby further underlining its feature as an 'allosteric hotspot'.

Next, we employed perturbation analysis to identify the allosteric communication pathway between the *Tco*PYK active site and the F16BP/F26BP effector binding pocket and its constituting critical residues. This unveiled an intriguing disparity with regards to the allosteric communication pathways in the enzyme's T and R state conformations (**Figure 4B**). In the TcoPYK T state tetramer, the allosteric pathways can be defined as 'intersubunit' since they run from the effector site in one subunit to the active site of a second subunit across the AA' interface. These pathways involve Arg311 and residues of the AA' interface, consistent with their roles in the 'rock and lock' mechanism during which conformational changes along intermonomer interfaces allow *Tco*PYK to transition from the T into the R state upon substrate/effector binding. Some of the pathways' critical residues are closely connected to the sdAb42 epitope: Ala21, Asn22, Ile350, and Cys351. This is corroborated by an additional investigation of the allosteric communication pathway within *Tco*PYK through entropy transfer analysis (**Hacisuleyman and Erman, 2017**), which identifies key residues involved in the T to R state transition (**Figure 4—figure supplement 1**), some of which are part of or closely connected to the sdAb42 epitope: Phe13, Pro15, Tyr143, Pro181, Cys183, and Val268. In contrast, the R state communication pathways link up the active and effector sites within individual monomers and can thus be considered as "intrasubunit". This results in a different set of critical residues, which no longer run past the sdAb42 epitope, hence explaining the lower ACI values of this site in the *Tco*PYK R state structure.

## The sdAb42 epitope is conserved in trypanosomatid pyruvate kinases

Given that trypanosomatid PYKs display a high degree of sequence identity (at least 70%; **Figure 5—figure supplement 1**), we assessed whether the 'allosteric hotspot' identified through perturbation analysis and the overlapping sdAb42 epitope would be conserved across *Trypanosoma* and *Leishmania*.

Mapping the degree of sequence conservation onto the structure of *Tco*PYK clearly illustrates that the sdAb42 epitope is well conserved among trypanosomatids (**Figure 5A**). Upon comparison with the sequences and structures of *T. brucei* and *L. mexicana* PYK (*Tbr*PYK and *Lme*PYK, respectively; two reference enzymes for studying the structure-function relationship of trypanosomatid PYKs), as little as three epitope residues differ: Lys43 (Gln43 in LmePYK), Val348 (Ala348 in *Lme*PYK), and Ile352 (Val352 and Leu352 in *Tbr*PYK and *Lme*PYK, respectively). The impact of these differences on binding energy was first assessed through an *in silico* ΔΔ G analysis, which predicts that the single

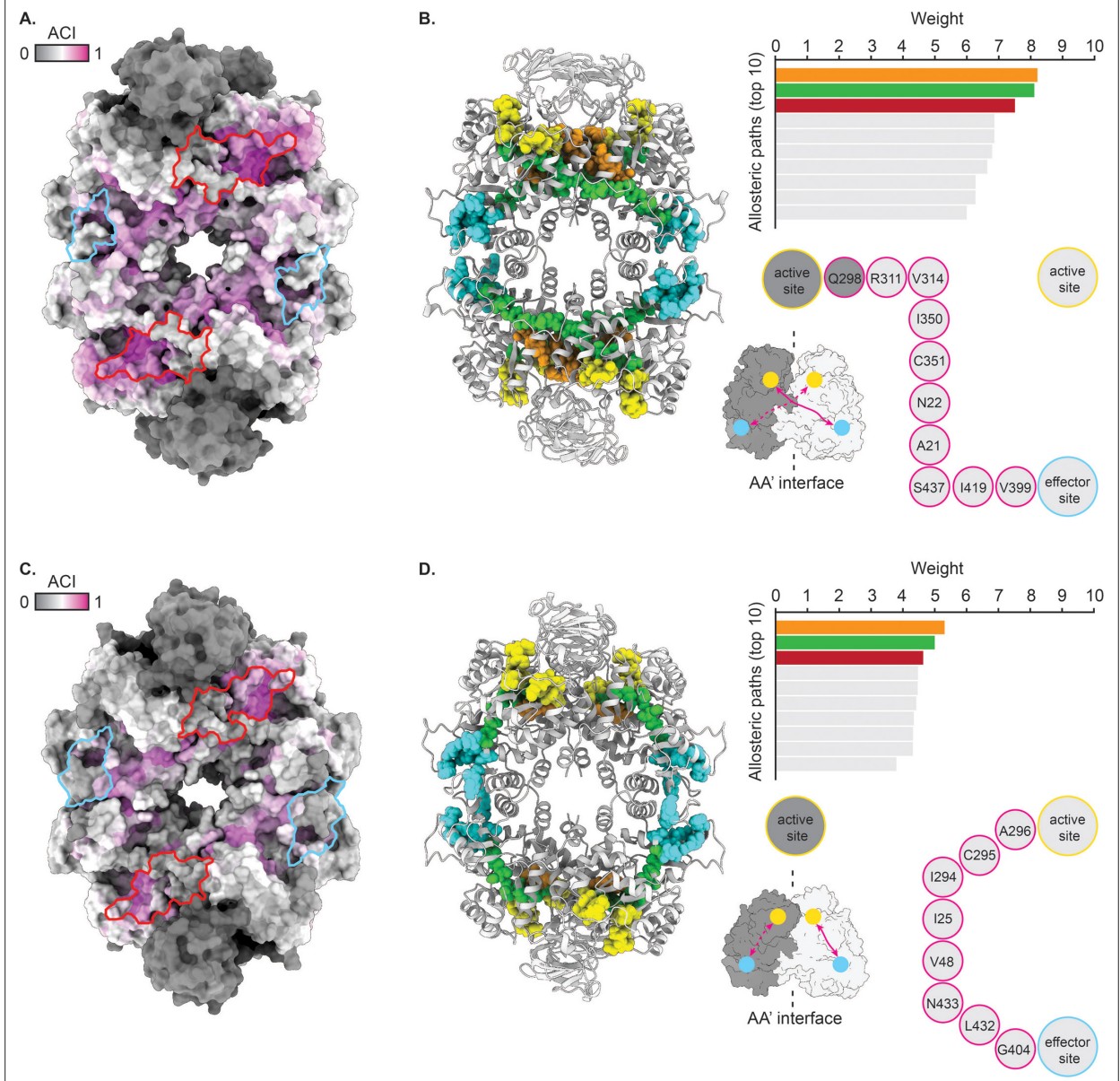

**Figure 4.** Perturbation analysis reveals distinct allosteric communication pathways in T and R state *Tco*PYK. (**A**, **C**) Surface representation of the *Tco*PYK tetramer in its T (**A**) and R state (**C**). The residues are color-coded according to their allosteric coupling intensity (ACI) values. The sdAb42 and effector molecule binding sites are delineated in red and cyan, respectively. (**B**, **D**) The left panel depicts a cartoon representation of the *Tco*PYK tetramer in its T (**B**) and R state (**D**) colored in light grey. The residues constituting the active site and effector binding site are shown in sphere representation and colored in yellow and cyan, respectively. The residues that form the top 3 allosteric communication paths (top right) are also shown in sphere representations and colored in orange, green, and dark red (the dark red and green paths overlap, which is why the dark red paths are not visible). The bottom right panel shows a schematic depiction of the inter- (**B**) and intrasubunit (**D**) allosteric communication pathways. The AA' dimer interface subunits are colored in dark and light grey, respectively, the active and effector binding sites are indicated by the yellow and cyan spheres, respectively, and the communication pathways are represented by the magenta arrows.

The online version of this article includes the following figure supplement(s) for figure 4:

**Figure supplement 1.** Entropy transfer and APOP analyses confirm the allosteric communication pathway and allosteric hotspot identified by perturbation analysis.

Ile352Val (corresponding *Tbr*PYK epitope) and triple Lys43Gln/Val348Ala/Ile352Leu (corresponding *Lme*PYK epitope) mutations are expected to have a negative impact on binding energy and thus affinity (*Table 2*). This is experimentally confirmed through the determination of the binding affinities of the different sdAb42 – trypanosomatid PYK complexes through isothermal titration calorimetry

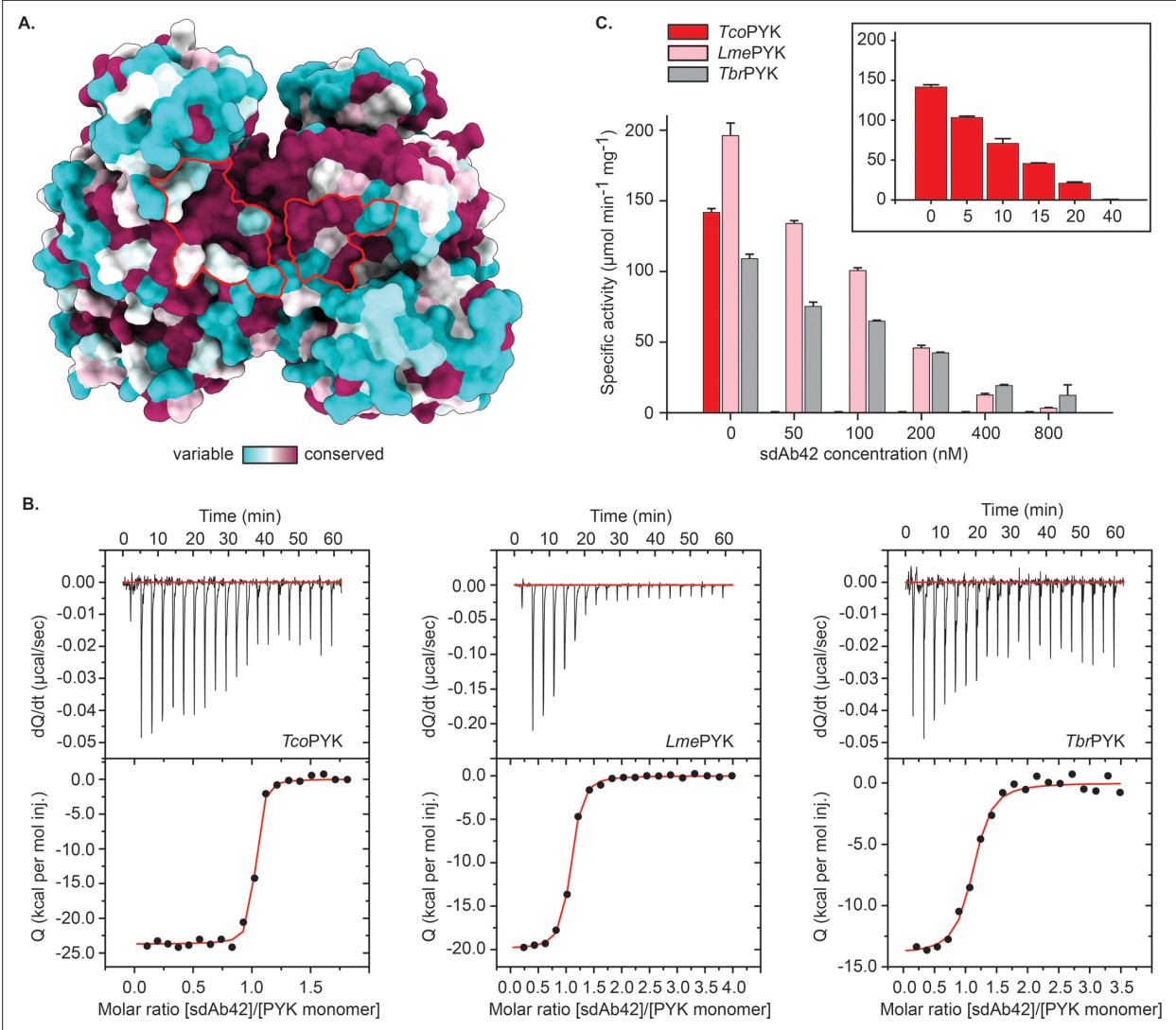

**Figure 5.** The sdAb42 epitope is conserved in trypanosomatid PYKs. (**A**) Surface representation of the *Tco*PYK AA' dimer interface monomers. The residues are color-coded according to their CONSURF conservation score based on a multiple sequence alignment of trypanosomatid PYKs. The sdAb42 epitope is delineated in red. (**B**) ITC measurements at 25 °C for the binding of sdAb42 to *Tco*PYK (left panel), *Lme*PYK (middle panel) and *Tbr*PYK (right panel). The top panels represent the thermograms in which the black lines depict the raw data. The bottom panels show the isotherms. The black dots display the experimental data points, and the red traces show the fit. (**C**) Effect of the addition of increasing concentrations of sdAb42 on the activity of *Tco*PYK (red bars), *Lme*PYK (pink bars), and *Tbr*PYK (grey bars) prior to addition of substrates and effectors. The results demonstrate that sdAb42 abrogates the activities of all tested trypanosomatid PYKs. The inset displays the effect of sdAb42 on *Tco*PYK activity at lower sdAb concentrations.

The online version of this article includes the following figure supplement(s) for figure 5:

**Figure supplement 1.** Amino acid sequence identities of trypanosomatid PYKs.

(ITC; *Figure 5B* and *Table 3*). As previously determined by surface plasmon resonance (SPR; *Pinto Torres et al., 2018*), sdAb42 binds *Tco*PYK with a high affinity in the low nM range ($K_D = 0.90 \pm 0.07$ nM), while the affinity of sdAb42 for *Tbr*PYK and *Lme*PYK is roughly 40-fold lower ($K_D$ values of $37.16 \pm 14.80$ nM and $42.54 \pm 10.81$ nM, respectively). Given that the three enzymes operate via the 'rock and lock' mechanism, it could be expected that the sdAb42 inhibition mechanism applies to all trypanosomatid PYKs. Indeed, the addition of sdAb42 to *Tbr*PYK and *Lme*PYK also impairs enzyme activity in a dose-dependent manner (*Figure 5C*). Compared to *Tco*PYK, higher sdAb42 concentrations are required to completely abolish enzyme activity, which is consistent with the lower affinity of sdAb42 for *Tbr*PYK and *Lme*PYK determined via ITC.

**Table 2.** Results of the *in silico* ΔΔG analysis.

The ΔΔG analysis was performed by uploading the sdAb42:*Tco*PYK structure to the mCSM-PPI2 (*Rodrigues et al., 2019*), mCSM-AB2 (*Myung et al., 2020b*), and mmCSM-AB (*Myung et al., 2020a*) servers and implementing the mutations of interest as specified by the author's instructions (https://biosig.lab.uq.edu.au/tools). Calculations were performed for those epitope residues that differ between *Tco*PYK, *Lme*PYK, and *Tbr*PYK. The single Ile352Val and triple Lys43Gln/Val348Ala/Ile352Leu mutants correspond to changes the *Tbr*PYK and *Lme*PYK epitopes, respectively.

**mCSM-PPI2**

| Wild-type | Position | Mutant | Distance to interface (Å) | ΔΔG (kcal mol$^{-1}$) | Affinity |
|-----------|----------|--------|---------------------------|------------------------|----------|
| Lys | 43 | Gln | 3.1 | 0.003 | Increasing |
| Val | 348 | Ala | 3.0 | -0.270 | Decreasing |
| Ile | 352 | Val | 3.6 | -0.326 | Decreasing |
| Ile | 253 | Leu | 3.6 | -0.282 | Decreasing |

**mCSM-AB2**

| Wild-type | Position | Mutant | Distance to interface (Å) | ΔΔG (kcal mol$^{-1}$) | Affinity |
|-----------|----------|--------|---------------------------|------------------------|----------|
| Lys | 43 | Gln | 3.1 | -0.13 | Decreasing |
| Val | 348 | Ala | 3.7 | -0.80 | Decreasing |
| Ile | 352 | Val | 3.6 | -0.29 | Decreasing |
| Ile | 253 | Leu | 3.6 | 0.34 | Increasing |

**mCSM-AB**

| Wild-type | Mutant | ΔΔG (kcal mol$^{-1}$) | Affinity |
|-----------|--------|------------------------|----------|
| Lys43, Val348, Ile352 | Gln43, Ala348, Leu352 | -1.32 | Decreasing |

## sdAb42 displays slow-binding inhibition kinetics against 'rocking and locking' trypanosomatid PYKs

The inhibition data shown in *Figures 2 and 5C* were performed by incubating the apo enzymes with sdAb42 prior to the addition of substrate and effector molecules. Based on our current working hypothesis and the relatively high affinity of sdAb42 toward trypanosomatid PYKs, this would readily lock the enzymes in their T state with very little to no chance of reverting back to the R state conformation. In a more realistic setting, the enzymes would be surrounded by substrate and effector molecules prior to their encounter with an exogeneous inhibitor. Consequently, the enzymes would be involved in their kinetic cycle by continuously transitioning between the T and R states. Hence, under such circumstances, we would expect that sdAb42 binding is delayed until the enzymes return to their T state conformation. Moreover, for inhibition to occur, this binding event must take place before the enzymes cycle back to the R state. In other words, enzyme inhibition under such circumstances would be expected to be less efficient.

To explore the inhibition behavior of sdAb42 in more detail, we performed the kinetic experiments by saturating the trypanosomatid PYKs with fixed effector concentrations prior to the addition of fixed, saturating substrate concentrations and increasing amounts of sdAb42. A careful inspection of the collected activity curves reveals the presence of non-linear features in the early phases, which is exacerbated with increasing inhibitor concentrations (especially pronounced for *Tco*PYK; *Figure 6*, top inset). Such non-linear behavior is typical for so-called 'slow-binding inhibition', which can indeed be

**Table 3.** Thermodynamic parameters determined via analysis of the ITC data.

All titrations were performed in triplicate at 25 °C (298.15 K).

| PYK-sdAb42 | N | $K_D$ (nM) | Δ G (kcal mol$^{-1}$) | Δ H (kcal mol$^{-1}$) | -T Δ S(kcal mol$^{-1}$) |
|------------|---|------------|------------------------|------------------------|--------------------------|
| *Tco*PYK | 1.00 ± 0.02 | 0.90 ± 0.07 | -12.34 ± 0.05 | -25.91 ± 1.99 | 13.57 ± 1.95 |
| *Lme*PYK | 1.00 ± 0.01 | 42.54 ± 10.81 | -10.07 ± 0.14 | -20.44 ± 0.50 | 10.37 ± 0.54 |
| *Tbr*PYK | 1.00 ± 0.06 | 37.16 ± 14.80 | -10.17 ± 0.25 | -13.83 ± 0.15 | 3.66 ± 0.14 |

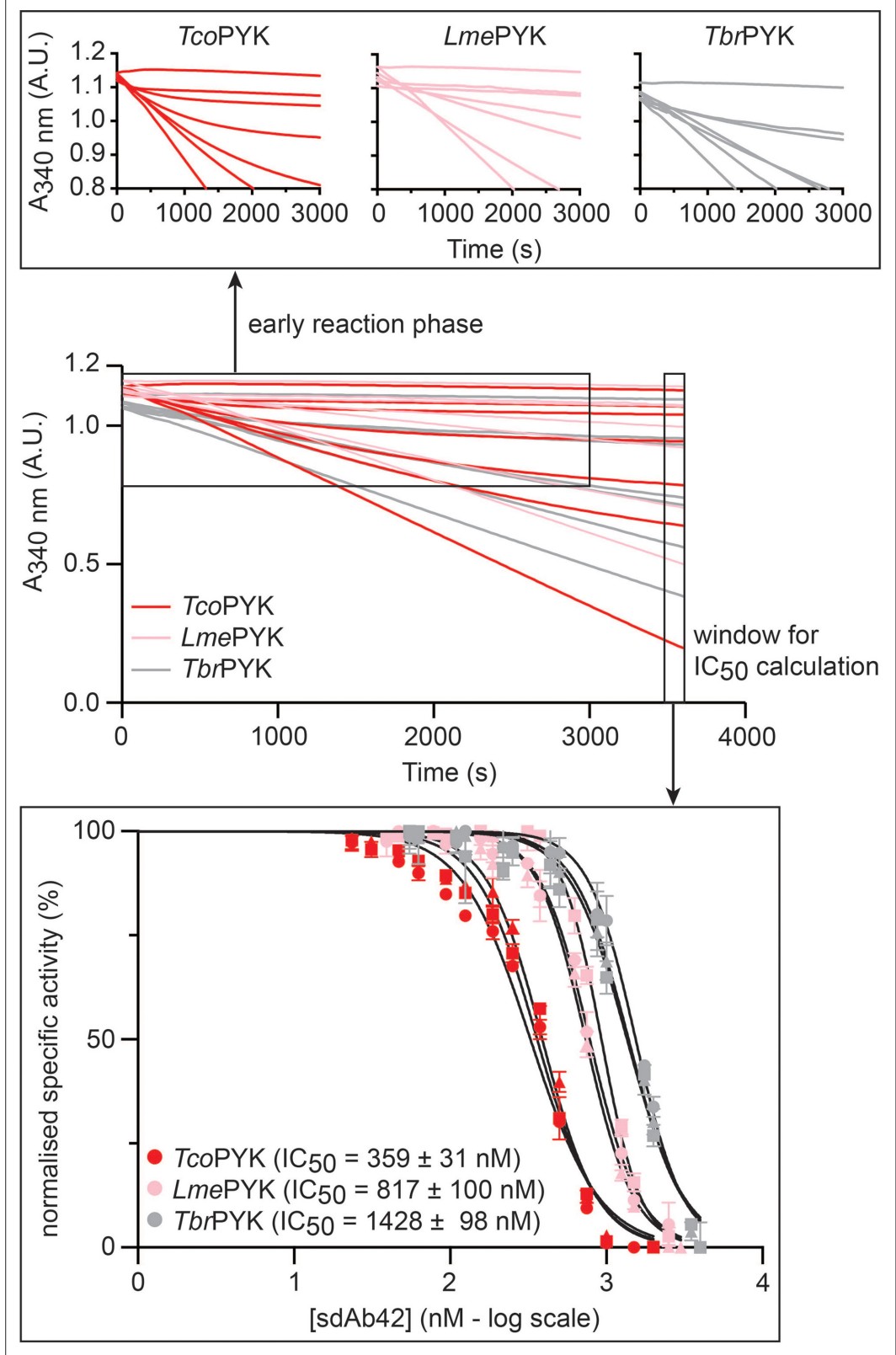

**Figure 6.** Slow binding inhibition kinetics. Full kinetic time traces for the reaction catalyzed by *Tco*PYK, *Lme*PYK, and *Tbr*PYK (red, pink, and grey traces, respectively) in the presence of fixed substrate/effector concentrations and increasing sdAb42 concentrations. Only a subset of the traces is shown for the sake of clarity. The following curves are shown (from bottom to top): *Tco*PYK (0.15 nM sdAb42, 500 nM sdAb42, 750 nM sdAb42, 1000 nM sdAb42,

*Figure 6 continued on next page*

*Figure 6 continued*

1500 nM sdAb42, 2000 nM sdAb42, no enzyme control), *Lme*PYK (5 nM sdAb42, 750 nM sdAb42, 1250 nM sdAb42, 1500 nM sdAb42, 2500 nM sdAb42, 3000 nM sdAb42, no enzyme control), and *Tbr*PYK (1 nM sdAb42, 1000 nM sdAb42, 1750 nM sdAb42, 2000 nM sdAb42, 3500 nM sdAb42, 4000 nM sdAb42, no enzyme control). The top inset shows a zoomed view of the activity curves to highlight the non-linear features of the traces. The bottom inset shows the $IC_{50}$ determination by only taking into account the rates at longer time ranges. The three independent inhibition assay replicates for each enzyme are indicated by the filled triangles, squares, and circles, respectively.

explained by the requirement of a *Tco*PYK R- to T-state transition prior to sdAb42-mediated enzyme inhibition, thereby supporting the above-mentioned hypothesis. While a full quantitative analysis of the slow-binding kinetics displayed by sdAb42 is beyond the scope of the current manuscript, the collected data sets allow for the determination of $IC_{50}$-values by only considering the rates at longer time ranges when the slow phase of inhibition has come into effect (*Figure 6*, bottom inset). In this way we find that sdAb42 inhibits *Tco*PYK with an $IC_{50}$ of ~350 nM, from which two observations can be made. First, the $IC_{50}$-value determined under these assay conditions is roughly 24-fold higher compared to the $IC_{50}$ measured under assay conditions in which *Tco*PYK is first incubated with sdAb42 prior to the addition of substrate and effector molecules (*Figure 2C*), which indeed confirms that enzyme inhibition is less efficient under these circumstances. Second, compared to *Tco*PYK, the $IC_{50}$-values for *Lme*PYK and *Tbr*PYK are increased two- to threefold, respectively, which is in accordance with the above-mentioned ITC experiments, with sdAb42 displaying a higher affinity towards *Tco*PYK compared to *Lme*PYK and *Tbr*PYK.

## The production of sdAb42 as an 'intrabody' induces a growth defect in a *T. brucei* model

Next, the impact of the sdAb42-mediated allosteric inhibition mechanism on parasite growth was investigated as a proof of concept. To this end, sdAb42 was used as an 'intrabody' by generating transgenic parasite lines capable of producing sdAb42 inside the parasite cytosol. The experimental design consisted of integrating a tetracycline (Tet)-inducible expression cassette in the 18 S rRNA locus, in which sdAb42 is C-terminally fused to mCherry such that cytosolic sdAb42 protein levels may be followed by fluorescence. The same approach was employed to generate a negative control line expressing sdAb BCII10. Since genetic engineering and culturing of *T. congolense* is notoriously difficult (*Awuah-Mensah et al., 2021*), we opted to perform our experiments in *T. brucei*.

The data presented in *Figure 7A* clearly demonstrate that (i) the intrabodies could successfully be produced upon Tet induction and that (ii) sdAb42 appears to induce a growth defect in a dose-dependent manner while BCII10 does not. A rapid loss of sdAb42 intrabody was noted, resulting in heterogenous in situ expression levels. To investigate the effect of sdAb levels on growth burden more thoroughly, a large-scale experiment was performed in which low and high intrabody expressing monoclonal lines were obtained by single cell sorting followed by growth curve analysis (*Figure 7B*). This reveals that increasing sdAb42 expression levels lead to larger growth defects, supporting a positive correlation between intrabody levels and impaired fitness (Pearson correlation, $r=0.95$; *Figure 7C*). As expected, BCII10 levels have no impact on parasite growth (Pearson correlation, $r=-0.72$), even at median fluorescence intensity (MFI) values that are significantly higher compared to those seen for the sdAb42 lines (*Figure 7B*). In addition, we observed that, while sdAb BCII10 levels remain stable over time, sdAb42 levels decrease rapidly (*Figure 7—figure supplement 1*). We suspect that this is the result of a parasite defense mechanism at the translational but not transcriptional level (sdAb42 transcript levels remain stable, data not shown) to specifically counter sdAb42 (but not BCII10) production. We interpret this finding as additional evidence for the detrimental effect of the sdAb42 intrabody on parasite growth.

## Discussion

The chemotherapeutic targeting of enzymes involved in energy metabolism has long shown promise in the battle against parasitic protists. Especially enzymes and transporters involved in glycolysis are of interest for drug development purposes for two main reasons. First, many parasitic protists heavily rely on glycolysis within their vertebrate hosts to sustain their energy metabolism: *Giardia* spp.

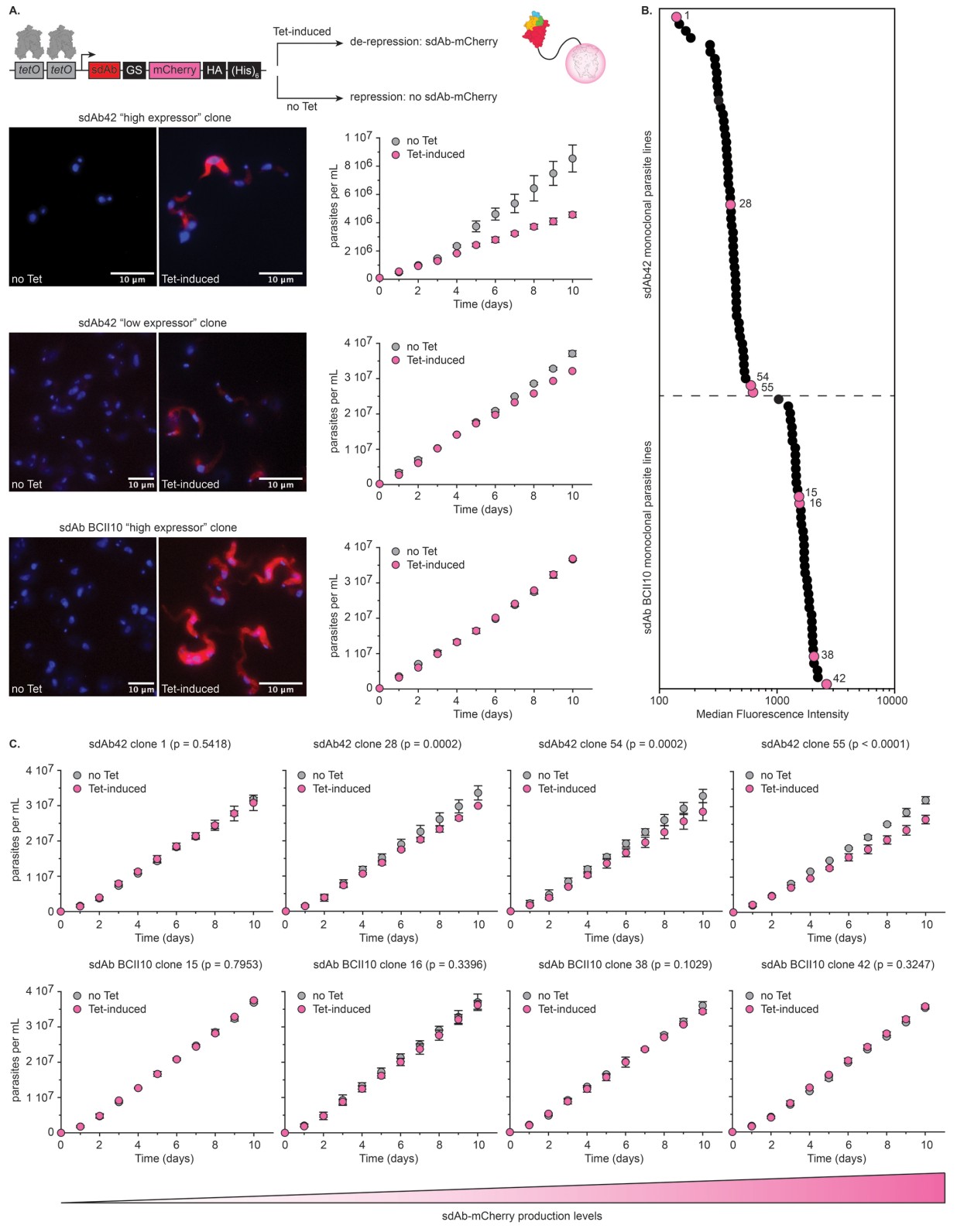

**Figure 7.** The intracellular production of sdAb42 generates a growth defect in *T. brucei*. (**A**) The top panel schematically depicts the principle underlying the tetracycline (Tet) controlled production of the sdAb-mCherry fusion protein. The panels in the bottom left show fluorescence microscopy pictures of transgenic trypanosomes prior to ('no Tet') and after Tet addition ('Tet-induced') for an sdAb42 'high expressor' clone, an sdAb42 'low expressor' clone, and an sdAb BCII10 'high expressor' clone. The panels in the bottom right show growth curves recorded for these clones under culture

*Figure 7 continued on next page*

*Figure 7 continued*

conditions without ('no Tet') and with Tet ('Tet-induced'). (**B**) Median fluorescence intensity (MFI) values for all obtained transgenic sdAb42 (55 clones) and sdAb BCII10 (42 clones) monoclonal parasite lines. Growth curves were measured for four selected sdAb42 and sdAb BCII10 clones (indicated by the pink spheres; sdAb42: clones 1, 28,54, and 55; sdAb BCII10: clones 15, 16, 38, and 42). (**C**) Results for the growth curves recorded for the clones highlighted in panel (**B**) under culture conditions without ('no Tet') and with Tet ('Tet-induced'). The clones were ranked from left to right based on the MFI values, which acts as a proxy for in situ intrabody levels (depicted by the gradient-colored triangle below the growth curves).

The online version of this article includes the following source data and figure supplement(s) for figure 7:

**Figure supplement 1.** Exponential decrease of sdAb42 protein levels over time.

**Figure supplement 1—source data 1.** This data set contains the original files of the full raw, uncropped, unedited blots that were employed to generate panel B.

**Figure supplement 1—source data 2.** This data set contains the original files of the full raw, uncropped, unedited blots that were employed to generate panel B including sample annotations.

trophozoites, *Trichomonas* spp., *Entamoeba* spp. trophozoites, *Plasmodium* spp. intraerythrocytic stages, *Leishmania* spp., *Trypanosoma* spp. BSFs (*Eisenthal and Cornish-Bowden, 1998*; *Upcroft and Upcroft, 2001*; *Saunders et al., 2010*; *Alam et al., 2014*). Second, the glycolytic enzymes and transporters are sufficiently different from their vertebrate host homologs such that they may be specifically targeted. Recent work on *T. brucei* and *Cryptosporidium parvum* demonstrates that achieving parasite killing by inhibiting glycolysis remains a viable avenue (*McNae et al., 2021*; *Khan et al., 2023*). Regarding trypanosomes, although the inhibition of glycolytic flux has been an area of intense research (*Bakker et al., 2000*; *Verlinde et al., 2001*; *Haanstra and Bakker, 2015*), there is a clear need for novel compound classes with novel modes of action and/or designed based on mechanistic insights in the target's structure-function relationship (*Field et al., 2017*; *De Rycker et al., 2018*). In this paper, we report the serendipitous discovery of a camelid sdAb (sdAb42) that allosterically inhibits the enzymatic activity of trypanosomatid PYKs.

sdAb42 was originally identified within a diagnostic context, in which its target *Tco*PYK was shown to be a reliable biomarker for the detection of active *T. congolense* infections (*Pinto Torres et al., 2018*). Besides possessing a diagnostic potential, our data now demonstrate that sdAb42 selectively and potently inhibits *Tco*PYK enzymatic activity. A thorough structural investigation reveals that this inhibition proceeds through an allosteric mechanism, in which sdAb42 selectively binds the enzyme's inactive T state, thereby 'locking' *Tco*PYK in a catalytically inactive conformation. An explanation for the latter is provided through ACI analysis (*Wang et al., 2020*), which suggests that the epitope targeted by sdAb42 is an 'allosteric hotspot' in the AA' intersubunit communication pathway required for the 'rocking and locking' of trypanosomatid PYKs. In other words, the binding of sdAb42 to this AA' intersubunit site is proposed to prevent *Tco*PYK from 'rocking and locking' into its active R state conformation. Interestingly, the location of this 'allosteric hotspot' is reminiscent of the binding site for free amino acids found in human PYKs (*Chaneton et al., 2012*; *Yuan et al., 2018*). Indeed, various amino acids have been shown to be allosteric regulators of human PYK activity. However, in contrast to human PYKs, trypanosomatid PYKs seem to be largely unresponsive to the allosteric regulation of enzyme activity by free amino acids (*Callens et al., 1991*). This resonates with the previous finding that PYKs from different species have evolved different allosteric strategies to regulate enzyme activity and that these differences could be exploited in drug discovery (*Morgan et al., 2014*).

The 'allosteric hotspot' targeted by sdAb42 is well conserved among trypanosomatids, which would suggest that this sdAb has the potential to inhibit other trypanosomatid PYKs. Indeed, the data presented here show that sdAb42 also blocks the enzymatic activity of *Tbr*PYK and *Lme*PYK (two reference enzymes for studying the structure-function relationship of trypanosomatid PYKs, *Morgan et al., 2010*; *Zhong et al., 2013*), albeit with a lower efficiency in comparison to *Tco*PYK. The explanation for these observations is two-fold. First, while most residues constituting the epitope are identical among the three investigated PYKs, the differences (Ile352Val and Lys43Gln/Val348Ala/Ile352Leu in *Tbr*PYK and *Lme*PYK, respectively) seem sufficient to cause the affinity to drop 40-fold. Second, the kinetic data recorded for experiments in which the trypanosomatid PYKs were first saturated with substrate and effector molecules prior to sdAb42 addition reveal that sdAb42 operates through a 'slow-binding inhibition' mechanism. Since the sdAb42 epitope is located relatively far away from both active and effector sites (~20 and ~40 Å, respectively), it seems highly unlikely that the observed 'slow-binding inhibition' kinetics are the result of a direct competition between sdAb42

and substrates and/or effectors. Instead, given that sdAb42 selectively binds and locks the enzyme's inactive T state, these data can be explained by the idea that sdAb42 can only bind to trypanosomatid PYKs after having undergone an R- to T-state transition. An additional observation in this context, is the 400-fold difference between the $K_D$ and $IC_{50}$ values. Although we currently do not have a mechanistic explanation, similar differences have been observed for the sdAb-mediated allosteric inhibition of other kinases (*Singh et al., 2022*).

In conclusion, the data demonstrate that sdAb42 inhibits three trypanosomatid PYKs through the same allosteric mechanism, of which the potency most likely depends on a combination of binding affinity and intrinsic enzyme dynamics. This probably also explains why the use of sdAb42 as an 'intrabody' in a *T. brucei* model generates a growth defect instead of completely killing off the parasite. Our results indicate that intracellular sdAb42 production impairs parasite growth in a dose-dependent manner, which we interpret as follows: increasingly higher intracellular sdAb42 levels lead to a higher degree of *Tbr*PYK complexation and inhibition, which in turn reduces the glycolytic flux (and/or lead to toxic accumulation of upstream glycolytic intermediates due to the lack of appropriate activity regulation mechanisms in enzymes such as hexokinase and PFK), thereby negatively impacting parasite fitness. The interplay between a lower binding affinity for *Tbr*PYK and the slow-binding inhibition mode necessitates that relatively high intracellular sdAb42 concentrations need to be reached to fully block all intracellular *Tbr*PYK activity, as suggested by the in vitro $IC_{50}$ measurements (*Tbr*PYK $IC_{50} \sim 1400$ nM). Earlier work by Albert and colleagues suggests that achieving trypanosome death through *Tbr*PYK inhibition would require an 88% reduction in the enzyme's $V_{max}$, thereby resulting in lowering the glycolytic flux below 50% (*Albert et al., 2005*). This is consistent with the observation that trypanosomes cannot survive more than 12 hr in a situation wherein their ATP synthesis flux is reduced by harvesting only 1 instead of 2 ATP molecules per glucose molecule (*Helfert et al., 2001*). Hence, it has been proposed that novel trypanosomatid PYK inhibitors should be used in concert with inhibitors of trypanosomatid glucose transporters and PFK to achieve synergistic effects (*Haanstra et al., 2011*). However, the high intracellular sdAb42 levels required to instill an 88% reduction in TbrPYK's $V_{max}$ are unlikely to be reached within this model system. Especially since we observed that the transgenic parasite lines appear to specifically counter sdAb42 production (but not of the BCII10 control). Interestingly, this counter-selection does not occur at the transcript, but at the protein level. While the exact mechanism remains unknown, trypanosomes are known to mainly regulate gene expression through extensive post-transcriptional mechanisms (*Clayton, 2019*).

Based on the presented results, we propose that sdAb42 may pinpoint a site of vulnerability on trypanosomatid PYKs that could potentially be exploited for the design of novel chemotherapeutics. Indeed, antibodies (or fragments thereof) are valuable drug discovery tools. Antibodies (and camelid sdAbs especially) are known for their ability to 'freeze out' specific conformations of highly dynamic antigens, thereby exposing target sites of interest, which could be exploited for rational drug design (*Lawson, 2012*; *Khamrui et al., 2013*; *van Dongen et al., 2019*). While the design of novel small molecules inspired on the sdAb42-mediated allosteric inhibition mechanism will require further investigation, the results presented here may provide a foundation to fuel such an endeavour.

## Materials and methods
### Cloning, recombinant protein production, and purification
All details concerning cloning, protein production and purification of *Tco*PYK have been previously described (*Pinto Torres et al., 2018*). Recombinant versions of PYKs from *T. brucei* and *L. mexicana* (*Tbr*PYK and *Lme*PYK, respectively) employed in this study were obtained with the same protocols. The two single-domain antibodies (sdAbs, previously termed Nb42 and Nb44) used in this work, were renamed as sdAb42 and sdAb44. All details related to their generation, identification, production, and purification, as well as details on the negative control sdAb BCII are described in *Pinto Torres et al., 2018*. The human PYK isoforms were produced and purified as described (*Yuan et al., 2018*).

### Enzymatic assays
The activity and enzyme kinetics of *Tco*PYK were measured and determined using a lactate dehydrogenase (LDH) coupled assay, as previously described (*Morgan et al., 2010*; *Pinto Torres et al., 2020*). To evaluate the inhibitory properties of sdAb42, sdAb44 and sdAb BCII10 on *Tco*PYK, an enzymatic

assay was performed with ADP and PEP concentrations of 2.5 mM and 5 mM, respectively. Fifty µl samples containing 88.8 nM (5 µg ml$^{-1}$) *Tco*PYK pre-mixed with varying sdAb molar ratios (TcoPYK-sdAb: 4:0, 4:1, 4:2, 4:4, 4:6) were incubated for at least 5 mins at 25 °C in buffer 1 (50 mM TEA buffer pH 7.2, 10 mM MgCl$_2$, 50 mM KCl, 10 mM F16BP) prior to the addition of 50 µl of buffer 2 (50 mM TEA buffer pH 7.2, 10 mM MgCl$_2$, 50 mM KCl, 5 mM ADP, 10 mM PEP, 2 mM NADH and 6.4 U LDH). Assay measurements were performed by following the decrease of the NADH absorbance at 340 nm, always ensuring that the PYK-catalyzed conversion of ADP and PEP to ATP and pyruvate is rate limiting. The activity was expressed as relative activity (in %), where values from samples containing only the enzyme (*Tco*PYK:sdAb at a ratio of 4:0) were taken as the 100% activity. To evaluate a possible inhibitory effect of sdAb42 and sdAb44 on human PYKs, an enzymatic assay was performed with the four human PYK isoforms: skeletal muscle M1 (M1PYK), muscle M2 (M2PYK), liver (LPYK), and red blood cells (RPYK). M1PYK, M2PYK, LPYK, and RPYK were diluted in PBS-CM (PBS without Mg$^{+2}$) to the following final concentrations: 10 µg ml$^{-1}$ (~170 nM for HM1PYK, HLPYK, and HRPYK) and 40 µg ml$^{-1}$ (~690 nM for HM2PYK). 25 µl of each enzyme was mixed with 25 µl sdAb42 (100 µg ml$^{-1}$, ~6.1 µM) or sdAb44 (100 µg ml$^{-1}$, ~6.4 µM). HPYKs-sdAbs (n=3 per sample) were allowed to incubate for 5 min at 25 °C prior to the addition of 50 µl buffer 3 (PBS pH 8.0, 2 mM PEP, 4 mM ADP, 1 mM NADH and 32 U ml$^{-1}$ LDH). Assay measurements were performed by following the decrease of the NADH absorbance at 340 nm. Slow binding kinetics assays were performed using a similar experimental set-up with the important difference that the trypanosomatid PYKs were saturated with fixed effector concentrations prior to the addition of fixed, saturating substrate concentrations and increasing amounts of sdAb42. Fifty µl samples containing 1.25 nM (*Tco*PYK and *Tbr*PYK) or 0.625 nM (*Lme*PYK) enzyme were prepared in buffer 1 (50 mM TEA buffer pH 7.2, 10 mM MgCl$_2$, 50 mM KCl, 10mM F16BP) and allowed to incubate for at least 5 min at 25 °C. Next, these samples were mixed with varying sdAb42 concentrations prepared in 50 µl buffer 2 (50 mM TEA buffer pH 7.2, 10 mM MgCl$_2$, 50 mM KCl, 5 mM ADP, 10 mM PEP, 2 mM NADH and 6.4 U LDH). Different sdAb42:PYK molar ratios were employed for *Tco*PYK (2000:1, 1500:1, 1000:1, 750:1, 375:1, 250:1, 167.5:1, 125:1, 84:1, 62.5:1, 42:1, 31:1, 22:1, 0.15:1), *Tbr*PYK (4000:1, 3500:1, 2000:1, 1750:1, 1000:1, 875:1, 500:1, 437.5:1, 250:1, 218.75:1, 125:1, 109.4:1, 62.5:1, 54.7:1, 0.62:1), and *Lme*PYK (3000:1, 2500:1, 1500:1, 1250:1, 750:1, 625:1; 375:1; 312.5:1, 187.5:1, 156.25:1, 93.8:1, 78.13:1, 47:1, 39:1, 4.7:1). Assay measurements were performed by following the decrease of the NADH absorbance at 340 nm for 3600 s at 25 °C. Three independent inhibition assays were performed per enzyme and each sdAb42 concentration was assessed in triplicate in each assay.

## Circular dichroism spectroscopy

Circular dichroism (CD) spectra were recorded on a J-715 spectropolarimeter (Jasco). Continuous scans were taken using a 1 mm cuvette at a scan rate of 50 nm min$^{-1}$ with a band width of 1.0 nm and a resolution of 0.5 nm. Six accumulations were taken at 25 °C in 20 mM Tris-HCl, 150 mM NaCl, pH 7.2 and a *Tco*PYK concentration of 0.2 mg ml$^{-1}$ (3.55 µM). The CD spectra for *Tco*PYK in complex with sdAb42 or sdAb44 were recorded after incubation of the sdAb42:*Tco*PYK (molar ratio of 4:4; four sdAb42 copies per *Tco*PYK tetramer) and sdAb44:*Tco*PYK (molar ratio of 2:4; two sdAb44 copies per *Tco*PYK tetramer) complexes for 30 min at 25 °C. The raw CD data (ellipticity $\theta$ in mdeg) were normalized for the protein concentration and the number of residues according to *Equation 1*, yielding the mean residue ellipticity ($[\theta]$ in deg cm$^2$ mol$^{-1}$), where MM, n, C, and l denote the molecular mass (Da), the number of amino acids, the concentration (mg ml$^{-1}$), and the cuvette path length (cm), respectively. Thermal unfolding experiments were performed by gradually increasing the temperature from 10 to 90°C at a constant rate of 1°C min$^{-1}$. To follow the change in $\alpha$-helicity, the mean residue ellipticity measured at 222 nm was plotted as a function of the temperature. The experimental data were fitted with the Boltzmann equation in Graphpad Prism to obtain the apparent melting temperature T$_{m,app}$.

$$[\theta] = \frac{\theta \cdot \mathrm{MM}}{(\mathrm{n} - 1) \cdot \mathrm{C} \cdot \mathrm{l}} \tag{1}$$

## Crystallization, data collection and processing, and structure determination

The sdAb42:*Tco*PYK complex (molar ratio of 4 sdAb42 copies per *Tco*PYK tetramer) was purified by gel filtration on a Superdex 200 16/60 column in 20 mM Tris, 150 mM NaCl, pH 8.0 as previously described (*Pinto Torres et al., 2018*). The complex was concentrated to 4.8 mg ml$^{-1}$ using a 50,000 molecular weight cut-off concentrator (Sartorius Vivaspin20). Crystallization conditions were screened manually using the hanging-drop vapor-diffusion method in 48-well plates (Hampton VDX greased) with drops consisting of 2 μl protein solution and 2 μl reservoir solution equilibrated against 150 μl reservoir solution. Commercial screens from Hampton Research (Crystal Screen, Crystal Screen 2, Crystal Screen Lite, Index, Crystal Screen Cryo), Molecular Dimensions (MIDAS, JCGS+), and Jena Bioscience (JBScreen Classic 1–10) were used for initial screening. The affinity tags of both *Tco*PYK and sdAb42 were retained for crystallization. The crystal plates were incubated at 20 °C. Diffraction-quality crystals of sdAb42:*Tco*PYK were obtained in JBScreen Classic 2 (Jena Bioscience) condition no. A4 (100 mM MES pH 6.5, 200 mM MgCl$_2$, 10% PEG 4000) and the crystals grew after approximately 10 days. For *Tco*PYK complexed by both sdAb42 and sulfate, diffraction quality crystals were obtained in PACT Premier (Molecular Dimensions) condition no. 2–32 (100 mM Bis-Tris propane pH 7.5, 200 mM sodium sulfate, 20% PEG 3350) and the crystals grew after a couple of weeks.

The sdAb42:*Tco*PYK and sdAb42:*Tco*PYK:sulfate crystals were cryocooled in liquid nitrogen with the addition of 25% (v/v) glycerol to the mother liquor as a cryoprotectant in 5% increments. Data sets for the sdAb42:*Tco*PYK and sdAb42:*Tco*PYK:sulfate crystals were collected at the SOLEIL synchrotron (Gif-Sur-Yvette, France) on the PROXIMA1 and PROXIMA2 beamlines, respectively. Both data sets were processed with XDSME (*Kabsch, 2010*; *Legrand, 2017*). The quality of the collected data sets was verified by close inspection of the XDS output files and through *phenix.xtriage* in the PHENIX package (*Liebschner et al., 2019*). Twinning tests were also performed by *phenix.xtriage*. Analysis of the unit cell contents was performed with the program MATTHEWS_COEF, which is part of the CCP4 package (*Winn et al., 2011*). The structures of sdAb42:*Tco*PYK and sdAb42:*Tco*PYK:sulfate were determined by molecular replacement with PHASER-MR (*McCoy et al., 2007*). The following search models were employed for molecular replacement: (i) six copies of the structure of the *Tco*PYK monomer (chain D, PDB ID: 6SU1) devoid of the B domain given its notorious flexibility, and (ii) six copies of an AlphaFold2 (*Jumper et al., 2021*; *Tunyasuvunakool et al., 2021*) model of sdAb42 (of which the CDR1 was removed due to poor pLDDT scores). This provided a single solution (top TFZ = 24.0 and top LLG = 9466.104). For both structures, refinement cycles using the maximum likelihood target function cycles of *phenix.refine* (*Liebschner et al., 2019*) were alternated with manual building using Coot (*Emsley and Cowtan, 2004*). The final resolution cut-off was determined through the paired refinement strategy (*Karplus and Diederichs, 2012*), which was performed on the PDB_REDO server (*Joosten et al., 2014*). The crystallographic data for the sdAb42:*Tco*PYK and sdAb42:*Tco*PYK:sulfate structures are summarized in *Table 4* and have been deposited in the PDB (PDB IDs: 8RTF and 8RVR, respectively). Molecular graphics and analyses were performed with UCSF ChimeraX (*Meng et al., 2023*).

## Perturbation, ΔΔG, transfer entropy, and APOP analyses

The perturbation analysis on the T and R state crystal structures of *Tco*PYK were performed as described by *Wang et al., 2020*. Tetramer structures of R state *Tco*PYK (*Pinto Torres et al., 2020*, PDB IDs: 6SU1 and 6SU2) and T state *Tco*PYK (this work) were uploaded to the Ohm server of the Dokholyan lab (https://dokhlab.med.psu.edu/ohm/) to identify (i) allosteric coupling intensities (ACI) of *Tco*PYK residues based on the active site and (ii) the allosteric pathways between the active and effector sites in both structures. The analysis was performed with the default server values (3.4 Å distance cutoff of contacts, 10,000 rounds of perturbation propagation, and $\alpha$ = 3.0). The ΔΔG analysis was performed by uploading the sdAb42:*Tco*PYK structure to the mCSM-PPI2 (*Rodrigues et al., 2019*), mCSM-AB2 (*Myung et al., 2020b*), and mmCSM-AB (*Myung et al., 2020a*) servers and implementing the mutations of interest as specified by the author's instructions (http://biosig.lab.uq.edu.au/tools). Transfer entropy (TE) is a concept introduced by *Schreiber, 2000* which quantifies the directional flow of information between two variables, capturing how the state of one variable predicts the future state of another beyond their shared past. The dGNM method introduced by *Hacisuleyman and Erman, 2017* integrates the transfer entropy concept with the Gaussian Network

**Table 4.** Data collection and refinement statistics.
Statistics for the highest resolution shell are shown in parentheses.

| | sdAb42:*Tco*PYK | sdAb42:*Tco*PYK:sulfate |
|---|---|---|
| **Data collection statistics** | | |
| Wavelength (Å) | 0.9792 | 0.9801 |
| Resolution range (Å) | 48.35 – 2.80 (2.97 – 2.80) | 48.34 – 3.19 (3.31 – 3.19) |
| Space group | 18 (P2$_1$2$_1$2) | 18 (P2$_1$2$_1$2) |
| a,b,c (Å) | 167.52, 170.81, 177.62 | 167.52, 168.42, 177.13 |
| α, β, γ (°) | 90, 90, 90 | 90, 90, 90 |
| Mosaicity (°) | 0.056 | 0.134 |
| Total number of measured reflections | 1 280 402 (202 776) | 1 152 904 (182 929) |
| Unique reflections | 125 473 (19 843) | 83 505 (13 176) |
| Multiplicity | 10.2 (10.2) | 13.8 (13.8) |
| Completeness (%) | 99.8 (99.1) | 99.8 (98.8) |
| $< I/\sigma(I) >$ | 12.53 (0.99) | 9.39 (1.10) |
| Wilson B-factor (Å$^2$) | 85.05 | 86.27 |
| R$_{meas}$ (%) | 13.9 (205.3) | 32.7 (240.70) |
| CC$_{1/2}$ (%) | 99.8 (61.2) | 99.5 (64.6) |
| A.U. contains | 6 sdAb42:*Tco*PYK complexes | 6 sdAb42:*Tco*PYK complexes |
| **Refinement statistics** | | |
| CC* | 1.00 (0.83) | 1.00 (0.83) |
| CC$_{work}$ | 0.95 (0.64) | 0.95 (0.74) |
| CC$_{free}$ | 0.96 (0.50) | 0.91 (0.59) |
| R$_{work}$ (%) | 23.73 (40.76) | 22.28 (36.16) |
| R$_{free}$ (%) | 27.40 (43.85) | 27.62 (38.51) |
| Number of non-hydrogen atoms | 25899 | 25988 |
| macromolecules | 25753 | 25757 |
| ligands | 60 | 108 |
| solvent | 86 | 123 |
| RMS bond lengths (Å) | 0.015 | 0.011 |
| RMS bond angles (°) | 2.01 | 1.49 |
| Ramachandran plot | | |
| favored (%) | 94.89 | 94.48 |
| allowed (%) | 5.08 | 5.46 |
| outliers (%) | 0.03 | 0.06 |
| Average B-factor (Å$^2$) | 105.07 | 112.42 |
| PDB ID | 8RTF | 8RVR |

Model (GNM) to map allosteric communication landscapes in proteins, by using the PDB structures only. By comparing the TE profiles of the protein in different states (e.g. R and T states), dGNM identifies significant shifts in communication networks. Analyzing the $\Delta\Delta T_{ij} = \left( \Delta T_{ij-Tstate} - \Delta T_{ij-Rstate} \right)$ and selecting the top 80% of significant changes highlights key residues and interactions driving functional state transitions. APOP (*Kumar et al., 2023*) is a method that predicts allosteric binding pockets

in proteins by analyzing how stiffening interactions across potential surface accessible pockets affects the protein's overall dynamics, combined with evaluating local hydrophobicity, to identify sites where ligand binding could influence protein behavior at distant locations.

## Sequence alignments

The amino acid sequences of kinetoplastid PYKs were obtained by performing a Protein BLAST search of the TriTrypDB (*Aslett et al., 2010*) using *Tco*PYK (Uniprot ID: G0UYF4) as the query sequence. A total of 17 kinetoplastid PYKs sequences (including *Tco*PYK) were employed to generate a multiple sequence alignment using MAFFT (*Katoh et al., 2002*).

## Isothermal titration calorimetry

The interactions between sdAb42 and trypanosomatid PYKs (*Tco*PYK, *Tbr*PYK and *Lme*PYK) were investigated by isothermal titration calorimetry (ITC) on a MicroCal PEAQ-ITC calorimeter system (Malvern Panalytical). In all experiments, the sdAb42 was titrated into the sample cell containing *Tco*PYK, *Tbr*PYK or *Lme*PYK. The following monomer concentrations were used for the different data sets: sdAb42 (10.0 µM) - *Tco*PYK (1.5 µM), sdAb42 (18.0 µM) - *Tbr*PYK (2.5 µM), and sdAb42 (79.5 µM) - *Lme*PYK (6.0 µM). All proteins were extensively dialyzed against the same buffer (20 mM Tris-HCl, 150 mM NaCl, pH 8.0) to exactly match buffer composition. Before being examined in the calorimeter, all samples were degassed for 10 min at a temperature close to the titration temperature (25 °C) to prevent long equilibration delays. Nineteen injections were used with a constant injection volume of 2.0 µl. The first injection was always 0.5 µl and its associated heat was never considered during data analysis. The reference power was set to 10 µcal s$^{-1}$ and a stirring speed of 750 rpm was used. An equilibrium delay of 360 s before the start of each measurement was employed, while a spacing of 180 s between each injection was used. Data analysis was performed with Origin 7.0 (OriginLab Corporation) and individual baselines for each peak were checked and, if applicable, manually modified for proper integration.

## Intrabody-expressing transgenic parasites

### Trypanosoma cultures

All assays were performed using the *T. brucei* Lister 427 single-marker cell line (Tb427sm) expressing a tetracycline repressor (TetR) and a T7 RNA polymerase (T7NRAP, *Wirtz et al., 1999*). Parasites were cultured in HMI-9 medium supplemented with 10% inactivated FBS and 2.5 µg ml$^{-1}$ G-418 (Life Technologies Europe) at 37 °C in a humidified atmosphere containing 5% $CO_2$ (G-418 was included for maintenance of TetR and T7RNAP). sdAb-expressing transgenic lines were subjected to an additional selection with 5 µg ml$^{-1}$ hygromycin B (HygB) (Sigma-Aldrich). Induction of the constructs was done by supplementing the medium with 0.5 µg ml$^{-1}$ tetracycline (Tet) (Takara Bio).

### Plasmids and transfection

Constructs encoding an sdAb-(GGGS)-(GGGGS)$_2$-mCherry fusion (in which the sdAb is either sdAb42 or control sdAb BCII10) equipped with a haemagglutinin and hexahistidine tag (hereafter referred to as intrabody) were synthesized and codon optimised for expression in *T. brucei brucei*. The commercially obtained constructs (Genscript) were then cloned by HiFi DNA Assembly (New England Biolabs) into the pLew100v5-HYG expression vector (Addgene, deposited by the George Cross lab, kindly provided to us by Prof. Isabel Roditi) that contains two *tet* operons and a *hygB* selection gene. 1 µg of the pLew100v5-HYG expression vector was digested overnight at 37 °C with HindIII-HF and BamHI-HF and loaded on a 1% agarose gel, after which the upper 6407 bp band was purified using a GeneJET Gel Extraction Kit (Thermo Fisher Scientific). Intrabody constructs were purified using the QIAquick PCR Purification Kit (Qiagen) and combined with the linearized pLew100v5-HYG expression vector in a 2:1 molar ratio together with 10 µl of HiFi DNA Assembly Master Mix and incubated for 1 hr at 50 °C. 1 µl of the ligation product was added to 50 µl of NEB 10-beta competent bacteria and transferred to an ice-cold 0.2 cm Gene Pulser cuvette (Bio-Rad) for electroporation (25 µF, 200 Ω and 2.5 kV) in a Bio-Rad Gene Pulser. Transformed bacteria were plated on LB-agar containing 100 µg ml$^{-1}$ ampicillin (Amp) and incubated overnight at 37 °C. Colonies were screened with a colony PCR targeting the sdAb, after which positive colonies were purified using a Nucleospin Plasmid miniprep kit (Macherey-Nagel) and sent for Sanger Sequencing on an Applied Biosystems 3730XL DNA

Analyzer (Neuromics Support Facility, University of Antwerp) and analysed using SnapGene. Bacteria containing the corresponding plasmids were grown in LB broth supplemented with 75 μg ml⁻¹ Amp (Sigma-Aldrich) under agitation at 37 °C for 24 hr, after which the plasmid was purified using the PureLink HiPure Plasmid Filter Midiprep Kit (Life Technologies Europe). After cutting 20 μg of plasmid DNA with a NotI-HF (New England Bioscience) and purification using the QIAquick PCR purification kit (QIAGEN), $4{\times}10^7$ Tb427sm cells were washed once in ice-cold cytomix (2 mM EDTA, 5 mM MgCl$_2$, 120 mM KCl, 0.15 mM CaCl$_2$, 10 mM K$_2$HPO$_4$, 25 mM HEPES, pH 7.6) and resuspended in 450 μl ice-cold cytomix in a 0.2 cm gap cuvette (Bio-Rad) together with the linearised DNA. Transfection was done using the Gene Pulser/MicroPulser Electroporation system (Bio-Rad) by applying two consecutive pulses, separated by a 10 s interval, at 1.5 kV, with 200 Ω resistance and 25 μF capacitance. Directly after transfection, cells were left to recover in 100 ml HMI-9 medium for 22 hr, after which HygB was added to select positive clones. Monoclonal lines were established utilising the microdrop method, with microscopic confirmation of the presence of a single cell. Two clones were initially selected for further analysis: one expressing the sdAb42 intrabody and one expressing the sdAb BCII10 intrabody. In a follow-up experiment, low and high intrabody expressing monoclonal lines were obtained by single cell sorting using a BD FACSMelody (BD Biosciences).

## Cumulative growth curve

Induced and non-induced cultures were seeded in duplicate in a 24-well plate at a density of $5{\times}10^5$ cells ml⁻¹ in 1 ml HMI-9 medium supplemented with HYG and G-418. For ten consecutive days, cells were counted using an improved Neubauer hematocytometer and subcultured in a 1:5 dilution. Based on the daily subculture and expansion rate, a cumulative growth curve was generated.

## Reverse Transcriptase Quantitative PCR (RT-qPCR)

After five days of induction, $5{\times}10^6$ cells from an exponential growth phase were washed once in PBS and RNA was isolated using the QIAGEN blood RNA isolation kit (Qiagen) following the manufacturer's recommendations. A one-Step SYBR green real-time PCR using primers targeting sdAb42 or sdAb BCII10 and the reference gene *TERT* was performed (primer sequences are provided in *Table 5*). Normalised mRNA expression was calculated by the $\Delta C_t$-method.

## Whole-cell lysates and western blotting

Whole-cell lysates were prepared five days after induction, by washing $5{\times}10^6$ cells in exponential growth phase twice with PBS and resuspending in 15 μl of 4% SDS. Lysates were diluted 1:1 with 2×Laemmli buffer (Bio-Rad) supplemented with 54 mg ml⁻¹ DTT. Standard western blots were performed by separating samples through gel electrophoresis and transferring to a polyvinylidene fluoride (PVDF) membrane. Intrabody expression was detected using an anti-HA HRP conjugated antibody (Genscript) and imaged employing chemiluminescence with the Clarity Western ECL substrate (Bio-Rad) in the Vilber Fusion FX imaging system.

## Flow cytometry

Intrabody expression was measured at days 2, 4, 7 and 9 of the parasite growth curve. For this, 1 ml of each culture was collected, centrifuged for 20 s at 20,000 × g and resuspended in 500 μl HMI-9 medium supplemented with HygB, G-418, Tet and 5% BD Via-Probe cell viability solution containing

**Table 5.** Primer sequences employed for the RT-PCR experiments.

| Name | Sequence |
| --- | --- |
| sdAb42-F | 5'-CAGAGACAACGCCAAGAACA-3' |
| sdAb42-R | 5'-ATCTCGGCCTGCACAGTAAT-3' |
| sdAb BCII-10-F | 5'-GGGTGGCCTCACATACTACG-3' |
| sdAb BCII-10-R | 5'-TCTGCAGAGTCACCGTGTTC-3' |
| TERT-F | 5'-GAGCGTGTGACTTCCGAAGG-3' |
| TERT-R | 5'-AGGAACTGTCACGGAGTTTGC-3' |

7-AAD (BD Biosciences). After a 15 min incubation at 37 °C, suspensions were centrifuged and resuspended in 500 µl HMI-9 medium. Intrabody expression levels were measured as mCherry fluorescence by flow cytometry using a BD FACSMelody (BD Biosciences). Data were analysed using the FlowLogic software version 7.3. The mCherry median fluorescence intensity (MFI) was determined for viable parasites within a selective FSC/SSC gate with exclusion of 7AAD + cells.

## Epifluorescence microscopy

Five days after induction, $5 \times 10^6$ exponentially growing cells were washed twice in PBS, resuspended in 100 µl PBS and spotted on poly-L-lysine coated coverslips. After a 30 min incubation at ambient temperature, cells were fixed for 30 min by adding 400 µl of 4% paraformaldehyde. Cover slips were washed three times with 1 ml PBS and mounted on microscopy slides with Fluoroshield mounting medium (Sigma-Aldrich). Images were taken with the Axio Observer Z1 (Zeiss) at ×60 and ×100 magnifications. Image analysis was done using ImageJ software version 1.52.

## Statistical analysis

All statistical analyses were performed in Prism version 8.4.1. For cumulative growth curves, a simple linear regression was modelled for each group, and the resulting slopes were compared using Brown-Forsythe ANOVA and Dunnett's T3 multiple comparison tests. Using non-linear regression, a one-phase decay model was fitted on the MFI and using the extra-sum-of-squares F test it was tested whether a single curve could fit both groups. Differences in MFI of the sdAb BCII10 clone was tested by means of a repeated measures one-way ANOVA with Tukey's multiple comparisons test.

## Acknowledgements

This work was supported by a Strategic Research Program Financing of the VUB (SRP95 to WV), the University of Ghent 'Bijzonder Onderzoeksfonds' (BOF.STG.2018.0009.01/01N01518/UGent BOF 'Startkrediet' to SM), the University of Antwerp 'Bijzonder Onderzoeksfonds' (41391 awarded to YG-JS), and the 'Fonds voor Wetenschappelijk Onderzoek – Vlaanderen' (FWO-Vlaanderen, G013518N to SM). MC. is a PhD fellow supported by FWO-Vlaanderen (1137622 N). YG-JS and GC participate in COST Action CA21111 (Onehealthdrugs). The authors wish to thank the staff of the SOLEIL synchrotrons PROXIMA 1 and 2 for outstanding beam line support.

## Additional information

### Funding

| Funder | Grant reference number | Author |
|---|---|---|
| Vrije Universiteit Brussel | SRP95 | Wim Versées |
| Universiteit Gent | BOF.STG.2018.0009.01/01N01518 | Stefan Magez |
| Universiteit Antwerpen | 41391 | Yann G-J Sterckx |
| Fonds Wetenschappelijk Onderzoek | G013518N | Stefan Magez |
| Fonds Wetenschappelijk Onderzoek | 1137622N | Mathieu Claes |
| COST Action CA21111 | CA21111 | Yann G-J Sterckx Guy Caljon |

The funders had no role in study design, data collection and interpretation, or the decision to submit the work for publication.

### Author contributions

Joar Esteban Pinto Torres, Conceptualization, Data curation, Formal analysis, Supervision, Validation, Investigation, Visualization, Methodology, Writing – original draft; Mathieu Claes, Meng Yuan, Formal

analysis, Validation, Investigation, Methodology, Writing – review and editing; Rik Hendrickx, Natalia Smiejkowska, Pieter Van Wielendaele, Hans De Winter, Serge Muyldermans, Validation, Investigation, Methodology, Writing – review and editing; Aysima Hacisuleyman, Formal analysis, Validation, Investigation, Visualization, Methodology, Writing – review and editing; Paul AM Michels, Malcolm D Walkinshaw, Supervision, Funding acquisition, Validation, Investigation, Methodology, Writing – review and editing; Wim Versées, Data curation, Formal analysis, Supervision, Funding acquisition, Validation, Investigation, Visualization, Methodology, Writing – review and editing; Guy Caljon, Conceptualization, Resources, Data curation, Software, Formal analysis, Supervision, Funding acquisition, Validation, Investigation, Visualization, Methodology, Writing – original draft, Writing – review and editing; Stefan Magez, Yann G-J Sterckx, Conceptualization, Resources, Data curation, Software, Formal analysis, Supervision, Funding acquisition, Validation, Investigation, Visualization, Methodology, Writing – original draft, Project administration, Writing – review and editing

### Author ORCIDs
Wim Versées ⓘ https://orcid.org/0000-0002-4695-696X
Guy Caljon ⓘ https://orcid.org/0000-0002-4870-3202
Yann G-J Sterckx ⓘ https://orcid.org/0000-0002-7420-0983

Reviewer #1 (Public review): https://doi.org/10.7554/eLife.100066.3.sa1
Reviewer #2 (Public review): https://doi.org/10.7554/eLife.100066.3.sa2
Reviewer #3 (Public review): https://doi.org/10.7554/eLife.100066.3.sa3
Author response https://doi.org/10.7554/eLife.100066.3.sa4

## Additional files

### Supplementary files
MDAR checklist

### Data availability
The crystallographic data for the sdAb42:TcoPYK and sdAb42:TcoPYK:sulfate structures have been deposited in the PDB (PDB IDs: 8RTF and 8RVR, respectively).

The following datasets were generated:

| Author(s) | Year | Dataset title | Dataset URL | Database and Identifier |
|---|---|---|---|---|
| Sterckx YG-J | 2024 | Crystal structure of Trypanosoma congolense pyruvate kinase in complex with a single-domain antibody (TcoPYK-sdAb42) | https://www.rcsb.org/structure/8RTF | RCSB Protein Data Bank, 8RTF |
| Sterckx YG-J | 2024 | Crystal structure of Trypanosoma congolense pyruvate kinase in complex with a single-domain antibody (TcoPYK-sdAb42) in the presence of sulfate | https://www.rcsb.org/structure/8RVR | RCSB Protein Data Bank, 8RVR |

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
