## [Editor Report · eLife Assessment]

This work presents **valuable** data demonstrating that a camelid single-domain antibody can selectively inhibit a key glycolytic enzyme in trypanosomes via an allosteric mechanism. The claim that this information can be exploited for the design of novel chemotherapeutics is **solid** but limited by the modest effects on parasite growth, as well as the lack of evidence for cellular target engagement in vivo.

---

## [Referee Report · Reviewer #1 (Public review)]

Summary:

The authors identified nanobodies that were specific for the trypanosomal enzyme pyruvate kinase in previous work seeking diagnostic tools. They have shown that a site involved in the allosteric regulation of the enzyme is targeted by the nanobody and using elegant structural approaches to pinpoint where binding occurs, opening the way to the design of small molecules that could also target this site.

Strengths:

The structural work shows the binding of a nanobody to a specific site on Trypanosoma congolense pyruvate kinase and provides a good explanation as to how binding inhibits enzyme activity. The authors go on to show that by expressing the nanobodies within the parasites they can get some inhibition of growth, which albeit rather weak, they provide a case on how this could point to targeting the same site with small molecules as potential trypanocidal drugs.

Weaknesses:

The impact on growth is rather marginal. Although explanations are offered on the reasons for that, including the high turnover rate of the expressed nanobody and the difficulty in achieving the high levels of inhibition of pyruvate kinase required to impact energy production sufficiently to kill parasites, this aspect of the work doesn't offer great support to developing small molecule inhibitors of the same site.

---

## [Referee Report · Reviewer #2 (Public review)]

Summary:

In this work, the authors show that the camelid single-chain antibody sdAb42 selectivity inhibits Trypanosome pyruvate kinase (PYK) but not human PYK. Through the determination of the crystal structure and biophysical experiments, the authors show that the nanobody binds to the inactive T-state of the enzyme, and in silico analysis shows that the binding site coincides with an allosteric hotspot, suggesting that nanobody binding may affect the enzyme active site. Binding to the T-state of the enzyme is further supported by non-linear inhibition kinetics. PYK is an important enzyme in the glycolytic pathway, and inhibition is likely to have an impact on organisms such a trypanosomes, that heavily rely on glycolysis for their energy production. The nanobody was generated against Trypanosoma congolense PYK, but for technical reasons the authors progressed to testing its impact on cell viability in *Trypanosoma brucei* brucei. First, they show that sdA42 is able to inhibit Tbb PYK, albeit with lower potency. Cell-based experiments next show that expression of sdA42 has a modest, and dose-dependent effect on the growth rate of Tbb. The authors conclude that their data indicates that targeting this allosteric site affects cell growth and is a valuable new option for the development of new chemotherapeutics for trypanosomatid diseases.

Strengths:

The work clearly shows that sdA42A inhibits Trypanosome and Leishmania PYK selectively, with no inhibition of the human orthologue. The crystal structure clearly identifies the binding site of the nanobody, and the accompanying analysis supports that the antibody acts as an allosteric inhibitor of PYK, by locking the enzyme in its apo state (T-state).

Weaknesses:

(1) The most impactful claim of this work is that sdAb42-mediated inhibition of PYK negatively affects parasite growth and that this presents an opportunity to develop novel chemotherapeutics for trypanosomatid diseases. For the following reasons I think this claim is not sufficiently supported:

- The authors do not provide evidence of target-engagement in cells, i.e. they do not show that sdA42A binds to, or inhibits, Tbb PYK in cells and/or do not provide a functional output consistent with PYK inhibition (e.g. effect on ATP production). Measuring the extent of target engagement and inhibition is important to draw conclusions from the modest effect on growth.

- The authors do not explore the selectivity of sdA42A in cells. Potentially sdA42A may cross-react with other proteins in cells, which would confound interpretation of the results.

- sdA42A only affects minor growth inhibition in Tbb. The growth defect is used as the main evidence to support targeting this site with chemotherapeutics, however based on the very modest effect on the parasites, one could reasonably claim that PYK is actually not a good drug target. The strongest effect on growth is seen for the high expressor clone in Figure 4a, however here the uninduced cells show an unusual profile, with a sudden increase in growth rate after 4 days, something that is not seen for any of the other control plots. This unexplained observation accentuates the growth difference between induced and uninduced, and the growth differences seen in all other experiments, including those with the highest expressors (clones 54 and 55) are much more modest. The loss of expression of sdA42A over time is presented as a reason for the limited effect, and used to further support the hypothesis that targeting the allosteric site is a suitable avenue for the development of new drugs. However, strong evidence for this is missing.

- For chemotherapeutic interventions to be possible, a ligandable site is required. There is no analysis provided of the antibody binding site to indicate that small molecule binding is indeed feasible.

(2) The authors comment on the modest growth inhibition, and refer to the need to achieve over 88% reduction in Vmax of PYK to see a strong effect, something that may or may not be achieved in the cell-based model (no target-engagement or functional readout provided). The slow binding model and switch of species are also raised as potential explanations. While these may be plausible explanations, they are not tested which leaves us with limited evidence to support targeting the allosteric site on PYK.

(3) The evidence to support an allosteric mechanism is derived from structural studies, including the in silico allosteric network predictions. Unfortunately, standard enzyme kinetics mode of inhibition studies are missing. Such studies could distinguish uncompetitive from non-competitive behaviour and strengthen the claim that sdAb42 locks the enzyme complex in the apo form.

(4) As general comment, the graphical representation of the data could be improved in line with recent recommendations: https://journals.plos.org/plosbiology/article?id=10.1371/journal.pbio.1002128, https://elifesciences.org/inside-elife/5114d8e9/webinar-report-transforming-data-visualisation-to-improve-transparency-and-reproducibility.

- Bar-charts for potency are ideally presented as dot plots, showing the individual data points, or box plots with datapoints shown.

- Images in Figure 7 show significant heterogeneity of nanobody expression, but the extent of this can not be gleaned from Figure 7B. It would be much better to use box plots or violin plots for each cell line on this figure panel. The same applies to Figure 10.

Comments on revision:

The authors have reduced the emphasis on the potential drug discovery applications. They are now referring to opportunities using a so called "chemo-superior" approach. This is not a commonly used term, and the newly added text seems to indicate that "chemo-superiors" target sites exposed by antibody binding, whereas the paper that the authors refer to (Lawson, 2012), defines "chemo-superiors" as small-molecules that induce similar effects to antibodies. I suggest removing the term "chemo-superior" altogether, as it has not been used since being coined in 2012, and instead simply point out the examples where antibodies have successfully informed small molecule design.

Unfortunately, the authors were unable to carry out additional experiments. Any experimental data to support their hypotheses as to why the observed growth defect is only marginal, and how the effect on growth could be increased, would have been very useful. As such, the evidence to support embarking on a drug discovery campaign for this allosteric site remains very limited.

The authors do provide some evidence of a druggable allosteric pocket, that partially overlaps with the antibody binding site, which is useful. However, I also ran the APOP tool on TcoPYK and it reveals 217 potential allosteric pockets all over the protein. The authors should provide the rank and APOP confidence score for the pocket that they have selected, to show that this is a high confidence allosteric pocket.

---

## [Referee Report · Reviewer #3 (Public review)]

Summary:

Out of the 20 Neglected Tropical Diseases (NTD) highlighted by the WHO, three are caused by members of the trypanosomatids, namely Leishmanaisis, Trypanosomiasis, and Chagas disease. Trypanosomal glycolytic enzymes including pyruvate kinase (PyK) have long been recognised as potential targets. In this important study, single-chain camelid antibodies have been developed as novel and potent inhibitors of PyK from the T, congolense. To gain structural insight into the mode of action, binding was further characterised by biophysical and structural methods, including crystal structure determination of the enzyme-nanobody complex. The results revealed a novel allosteric mechanism/pathway with significant potential for the future development of novel drugs targeting allosteric and/or cryptic binding sites.

Strengths:

This paper covers an important area of science towards the development of novel therapies for three of the Neglected Tropical Diseases. The manuscript is very clearly written with excellent graphics making it accessible to a wide readership beyond experts. Particular strengths are the wide range of experimental and computational techniques applied to an important biological problem. The use of nanobodies in all areas from biophysical binding experiments and X-ray crystallography to in-vivo studies is particularly impressive. This is likely to inspire researchers from many areas to consider the use of nanobodies in their fields.

Weaknesses:

There is no particular weakness, but I think the computational analysis of allostery, which basically relies on a single server could have been more detailed.

---

## [Author Response]

The following is the authors’ response to the original reviews.

**eLife assessment**
“This work presents valuable data demonstrating that a camelid single-domain antibody can selectively inhibit a key glycolytic enzyme in trypanosomes via an allosteric mechanism. The claim that this information can be exploited for the design of novel chemotherapeutics is incomplete and limited by the modest effects on parasite growth, as well as the lack of evidence for cellular target engagement in vivo.”

We agree with this assessment. In this re-worked version, we implemented the textual changes suggested by the reviewers and performed additional in silico work. The reviewers also presented valuable suggestions for additional experiments. However, we currently don’t have dedicated hands and funding for this project, which renders it impossible for us to perform additional “wet lab” experiments at this stage. We have thus not included new experimental “wet lab” data. Finally, the claim that our results may be exploited for the design of novel chemotherapeutics perhaps came across stronger than we intended to. We still believe our findings indicate a potential for such an endeavor, but this clearly requires further investigation and experimental evidence. We have softened this statement by removing it from the abstract and have edited the discussion to end as follows.

“Based on the presented results, we propose that sdAb42 may pinpoint a site of vulnerability on trypanosomatid PYKs that could potentially be exploited for the design of novel chemotherapeutics. Indeed, antibodies (or fragments thereof) are valuable drug discovery tools. Antibodies (and camelid sdAbs especially) are known for their ability to "freeze out" specific conformations of highly dynamic antigens, thereby exposing target sites of interest, which could be exploited for rational drug design (the development of so-called "chemo-superiors", (Lawson, 2012; Khamrui et al., 2013; van Dongen et al., 2019)). While the design of a "chemo-superior" inspired on the sdAb42-mediated allosteric inhibition mechanism will require further investigation, the results presented here provide a foundation to fuel such an endeavour.”

**REVIEWER 1:**
Summary:The authors identified nanobodies that were specific for the trypanosomal enzyme pyruvate kinase in previous work seeking diagnostic tools. They have shown that a site involved in the allosteric regulation of the enzyme is targeted by the nanobody and using elegant structural approaches to pinpoint where binding occurs, opening the way to the design of small molecules that could also target this site.Strengths:The structural work shows the binding of a nanobody to a specific site on Trypanosoma congolense pyruvate kinase and provides a good explanation as to how binding inhibits enzyme activity. The authors go on to show that by expressing the nanobodies within the parasites they can get some inhibition of growth, which albeit rather weak, they provide a case on how this could point to targeting the same site with small molecules as potential trypanocidal drugs.Weaknesses:The impact on growth is rather marginal. Although explanations are offered on the reasons for that, including the high turnover rate of the expressed nanobody and the difficulty in achieving the high levels of inhibition of pyruvate kinase required to impact energy production sufficiently to kill parasites, this aspect of the work doesn't offer great support to developing small molecule inhibitors of the same site.
**Recommendations for authors:**
Generally, the paper is very well written and the figures and their legends are clear.Comment 1.1: I thought the Introduction could give more focus to the need for new drugs for veterinary trypanosomiasis. The reality is that with fexinidazole now available and acoziborole soon to be available, with <1,000 cases of human African trypanosomiasis in each of the last five years, the case for needing new drugs is difficult to make. For Animal trypanosomiasis, however, the need for novel drugs is much more pressing.

We agree with this comment and have included an additional section in the Introduction’s second paragraph, which reads as follows.

“Hence, there is a need for alternative compounds, preferably with novel modes of action and/or designed based on mechanistic insights of the target’s structure-function relationship (Field et al., 2017; De Rycker et al., 2018). This need is especially pressing for AAT, which strongly impedes sustainable livestock rearing in Sub-Saharan Africa. AAT results in drastic reductions of draft power, meat, and milk production by the infected animals (small and large ruminants), and its control relies mainly on vector control and chemotherapy, with only few drugs currently available. The lack of routine field diagnosis has resulted in the misuse of trypanocidal drugs, thereby accelerating the rise of parasite resistance and further exacerbating the problem (Richards et al., 2021). As such, AAT-inflicted annual losses are estimated at around $5 billion (and the necessity to invest another $30 million each year to control AAT through chemotherapy), thereby having a devastating impact on the socio-economic development of Sub-Saharan Africa (Fetene et al., 2021). In contrast, HAT is perceived as a minor threat as it has reached a post-elimination phase as a public health problem with less than 1,000 yearly documented cases (Franco et al., 2022). In addition, new and effective drugs for HAT treatment have recently become available (De Rycker et al., 2023). HAT control currently relies on case detection and treatment, and vector control (Büscher et al., 2017).”

Comment 1.2: A few pedantic things can be tidied up too, for example on line 61 it is stated tsetse flies are part of the life cycle for all trypanosomes while some veterinary species e.g. T. evansi and some T.vivax strains use other biting flies for transmission. I'd also add in the Introduction that pyruvate kinase is not a glycosomal enzyme it is covered in the legend to figure 1 but I think it is quite important to clarify in the Introduction too so as to assure readers aren't wondering if "intrabodies" can get targeted there.

We agree with this comment and have included an additional section in the Introduction’s third paragraph to expand on the life cycles of African trypanosomes, which reads as follows.

“African trypanosomes are extracellular parasites that have a bipartite life cycle involving insect vectors and mammals as hosts (Radwanska et al., 2018). Most HAT (*T. brucei* gambiense and T. b. rhodesiense) and AAT (T. b. brucei and T. congolense) causing trypanosomes are uniquely vectored by tsetse flies (Glossina spp.) and are confined to Sub-Saharan Africa. T. b. evansi and T. vivax (both causative agents of AAT) have expanded beyond the tsetse belt due to their ability to be mechanically transmitted by a variety of biting flies (Glossina, Stomoxys, and Tabanus spp.). Finally, T. b. equiperdum infects equids and represents an exception as it is transmitted directly from animal to animal through sexual contact.”

The introduction now also explicitly mentions that pyruvate kinase is not a glycosomal enzyme.

Comment 1.3: The introduction would also be a good place to include some more information on what is known about the allosteric effectors of pyruvate kinase in trypanosomes, and emphasize where gaps in knowledge exist too.

We agree with this comment and have included an additional section in the Introduction’s third paragraph, which reads as follows.

“Pyruvate kinase (PYK) represents another attractive glycolytic target. This non-glycosomal enzyme catalyses the last step of the glycolysis (the irreversible conversion of phosphoenolpyruvate (PEP) to pyruvate; Figure 1A). The importance of this reaction is two-fold: (i) the generation of ATP through the transfer of a phosphoryl group from PEP to ADP and (ii) the formation of pyruvate, a crucial metabolite of the central metabolism. Like most PYKs, trypanosomatid PYKs are homotetramers. The PYK monomer is a ∼55 kDa protein organized into four domains termed ’N’, ’A’, ’B’, and ’C’ (Figure 1B). The A domain constitutes the largest part of the PYK monomer and is characterized by an (𝛼/𝛽)8-TIM barrel fold that contains the active site. Together with the N-terminal domain, it is also involved in the formation of the PYK tetramer AA’ dimer interfaces. The B domain is known as the flexible ’lid’ domain that shields the active site during enzyme-mediated phosphotransfer. Finally, the C domain harbors the binding pocket for allosteric effectors and stabilizes the PYK tetramer by formation of CC’ dimer interfaces. Because of their role in ATP production and distribution of fluxes into different metabolic branches, the activity of trypanosomatid PYKs is tightly regulated through an allosteric mechanism known as the "rock and lock" model (Morgan et al., 2010, 2014; Pinto Torres et al., 2020). In this model (which is detailed in Figure 1C), the binding of substrates and/or effectors (and analogs thereof) to the active and effector sites, respectively, trigger a conformational change from the enzymatically inactive T state to the catalytically active R state. Known effector molecules for trypanosomatid PYKs are fructose 2,6-bisphosphate (F26BP), fructose 1,6-bisphosphate (F16BP) and sulfate (SO_4_^2-^), with F26BP being the most potent one (van Schaftingen et al., 1985; Callens and Opperdoes, 1992; Ernest et al., 1994; Tulloch et al., 2008). Interestingly, trypanosomatid PYKs seem to be largely unresponsive to the allosteric regulation of enzyme activity by free amino acids (Callens et al., 1991), which contrasts with human PYKs (Chaneton et al., 2012; Yuan et al., 2018). Known trypanosomatid PYK inhibitors impair enzymatic activity through occupation of the PYK active site (Morgan et al., 2011).”

In the Results, although I am not qualified to analyse the structural data in detail I am confident in the ability of the authors to do so.

Comment 1.4: Differences in nanobody binding kinetics to the T. congolense enzyme when compared to *T. brucei* and Leishmania enzymes are attributed to the relatively few amino acid differences in those sites. It is desirable to test site-directed mutagenesis of those residues.

This is a highly valuable suggestion from the reviewer. However, we currently don’t have dedicated hands and funding for this project, which renders it impossible for us to perform additional experiments at this stage.

Comment 1.5: In the section on slow-binding inhibition kinetics (lines 194-220) I found it difficult to follow whether it was just the R<>T transition that slowed nanobody inhibition, or whether competition with effectors at the site would also impact on those inhibition kinetics. Can this be clarified?

Since the sdAb42 epitope is located relatively far away from both active and effector sites (~20 and ~40 Å, respectively), it seems highly unlikely the observed “slow-binding inhibition” kinetics are the result of a competition between sdAb42 on one hand and substrates and/or effectors on the other for enzyme binding. Instead, given that sdAb42 selectively binds and locks the enzyme’s inactive T state, these data can be explained by the idea that sdAb42 can only bind to trypanosomatid PYKs after having undergone an R- to T-state transition. To clarify this matter, we slightly reformulated the discussion as indicated below. We also included a small discussion on the observation that there is a 400-fold difference between the Kd and the IC50.

“Since the sdAb42 epitope is located relatively far away from both active and effector sites (~20 and ~40 Å, respectively), it seems highly unlikely that the observed “slow-binding inhibition” kinetics are the result of a direct competition between sdAb42 and substrates and/or effectors. Instead, given that sdAb42 selectively binds and locks the enzyme’s inactive T state, these data can be explained by the idea that sdAb42 can only bind to trypanosomatid PYKs after having undergone an R- to T-state transition. An additional observation in this context, is the 400-fold difference between the K_D_ and IC_50_ values. Although we currently do not have a mechanistic explanation, similar differences have been observed for the sdAb-mediated allosteric inhibition of other kinases (Singh et al., 2022).”

For the intrabody expression work, the reference cited on line 230 actually points to a growing ability to genetically modify *T. congolense*. However, it is justifiable to work on *T. brucei* given the much wider availability and advanced status of the genetic tools.

The growth inhibition data shown in Figure 7 is weak, albeit significant and the case is made as to why that might be.

Comment 1.6: The authors do point to the fact that inhibiting other parts of the glycolytic pathway might be helpful in getting a better growth inhibitory effect. It would be useful, in this regard, to test the ability of the PFK inhibitors in the Macnae et al. paper in the intrabody expressing line, and possibly other inhibitors e.g. 2-deoxy-D-glucose to see if these combinations do have the desired impacts. Also, looking at the metabolome of the intrabody expressors under induction could also give some further insights into changes in flux (although perhaps not on its own given the weak effects on the growth seen).

This is a highly valuable suggestion from the reviewer. However, we currently don’t have dedicated hands and funding for this project, which renders it impossible for us to perform additional experiments at this stage. We would like to point out that, in our experience, studying the effect of enzyme inhibition on the metabolome is usually only useful shortly after adding the onset of inhibition. The system adapts to the lowered flux and relevant changes are mostly transient. Since the induced expression of sdAb42 is almost certainly slow, we expect the metabolic changes will be minimal.

**REVIEWER 2:**
Summary:In this work, the authors show that the camelid single-chain antibody sdAb42 selectivity inhibits Trypanosome pyruvate kinase (PYK) but not human PYK. Through the determination of the crystal structure and biophysical experiments, the authors show that the nanobody binds to the inactive T-state of the enzyme, and in silico analysis shows that the binding site coincides with an allosteric hotspot, suggesting that nanobody binding may affect the enzyme active site. Binding to the T-state of the enzyme is further supported by non-linear inhibition kinetics. PYK is an important enzyme in the glycolytic pathway, and inhibition is likely to have an impact on organisms such a trypanosomes, that heavily rely on glycolysis for their energy production. The nanobody was generated against Trypanosoma congolense PYK, but for technical reasons the authors progressed to testing its impact on cell viability in *Trypanosoma brucei* brucei. First, they show that sdA42 is able to inhibit Tbb PYK, albeit with lower potency. Cell-based experiments next show that expression of sdA42 has a modest, and dose-dependent effect on the growth rate of Tbb. The authors conclude that their data indicates that targeting this allosteric site affects cell growth and is a valuable new option for the development of new chemotherapeutics for trypanosomatid diseases.Strengths:The work clearly shows that sdA42A inhibits Trypanosome and Leishmania PYK selectively, with no inhibition of the human orthologue. The crystal structure clearly identifies the binding site of the nanobody, and the accompanying analysis supports that the antibody acts as an allosteric inhibitor of PYK, by locking the enzyme in its apo state (T-state).Weaknesses:(1) The most impactful claim of this work is that sdAb42-mediated inhibition of PYK negatively affects parasite growth and that this presents an opportunity to develop novel chemotherapeutics for trypanosomatid diseases. For the following reasons I think this claim is not sufficiently supported:Comment 2.1: The authors do not provide evidence of target-engagement in cells, i.e. they do not show that sdA42A binds to, or inhibits, Tbb PYK in cells and/or do not provide a functional output consistent with PYK inhibition (e.g. effect on ATP production). Measuring the extent of target engagement and inhibition is important to draw conclusions from the modest effect on growth.The authors do not explore the selectivity of sdA42A in cells. Potentially sdA42A may cross-react with other proteins in cells, which would confound interpretation of the results.

We understand the reviewer’s concern. While it is theoretically possible that sdAb42 may non-specifically (cross-)react with other proteins in the cell, this would be highly unlikely based on the following arguments. First, many studies have employed sdAbs as intrabodies and reported on specific sdAb-mediated effects (outstanding reviews on the topic are Cheloha et al. (PMID 32868455) and Soetens et al. (PMID 33322697)). Second, it has been demonstrated that selecting sdAbs from an immune library through phage display or “bacteriomatch” (a bacterial system similar to yeast two hybrid) yields highly similar results (Pellis et al., PMID 22583807), thereby indicating that sdAbs interact specifically with their target antigens in an intracellular environment. Third, we identified TcoPYK as the target for sdAb42 by employing sdAb42 as bait in a pull-down from a parasite whole cell lysate (Pinto Torres et al., PMID 29899344). The pull-down fractions were analysed by SDS-PAGE and we observed a clear prominent band, which was further analysed by mass spectrometry and revealed TcoPYK as the target with great certainty. Even though the affinity of sdAb42 for TbrPYK is lower, it still remains high (nM affinity) and we expect it to bind TbrPYK with high specificity.

Regarding measuring the effect on ATP production, we would like to state that such experiments are not obvious. Instead of measuring ATP levels, one should measure ATP turnover as ATP levels may not necessarily be decreased. The latter was observed to be the case for the specific inhibition of trypanosomal PFK (Nare et al. PMID 36864883). The specific trypanosomal PFK inhibitor inhibits motility (and growth) of T. congolense IL3000 at concentrations that only slightly affect ATP levels. One could think of repeating the sdAb42 experiments in a T. congolense model. However, T. congolense BSF metabolism is more complicated than that of *T. brucei* BSF. First, the T. congolense glucose metabolic network is more expanded, allowing a lower glucose consumption rate to produce ATP and metabolites for growth. Second, pyruvate is not excreted but further metabolised, in part in the mitochondrion. Steketee et al. (PMID 34310651) have shown that T. congolense also takes up pyruvate from the medium. One can thus check if (increased) external pyruvate (partially) rescues the growth inhibition by sdAb42. It will not provide proof, but maybe an indication. As mentioned above, we are currently unable to perform such additional experiments due to lack of dedicated hands and funding.

Comment 2.2: sdA42A only affects minor growth inhibition in Tbb. The growth defect is used as the main evidence to support targeting this site with chemotherapeutics, however based on the very modest effect on the parasites, one could reasonably claim that PYK is actually not a good drug target. The strongest effect on growth is seen for the high expressor clone in Figure 4a, however here the uninduced cells show an unusual profile, with a sudden increase in growth rate after 4 days, something that is not seen for any of the other control plots. This unexplained observation accentuates the growth difference between induced and uninduced, and the growth differences seen in all other experiments, including those with the highest expressors (clones 54 and 55) are much more modest. The loss of expression of sdA42A over time is presented as a reason for the limited effect, and used to further support the hypothesis that targeting the allosteric site is a suitable avenue for the development of new drugs. However, strong evidence for this is missing.

We agree that the growth effect of sdAb42 expression is modest, and we have provided several explanations as to why this could be the case. In addition, as mentioned at the start of this rebuttal, the claim that our results may be exploited for the design of novel chemotherapeutics was perhaps expressed stronger than we intended to. We still believe our findings indicate a potential for such an endeavor, but this clearly requires further investigation and experimental evidence as mentioned by the reviewer.

We, however, disagree that PYK would not be a good drug target. Its potential to serve as a drug target is related to its fundamentally important role in trypanosomal glycolysis and not to the features of sdAb42. Steketee et al. (PMID 34310651) have shown that glycolysis is essential for *T. congolense* BSF, despite a lower glycolytic flux than in *T. brucei* BSF. The *T. congolense* glucose metabolic network is more expanded, allowing a lower glucose consumption rate to produce ATP and metabolites for growth. Also here, PYK is thus almost certainly essential and from that perspective a good drug target.

Comment 2.3: For chemotherapeutic interventions to be possible, a ligandable site is required. There is no analysis provided of the antibody binding site to indicate that small molecule binding is indeed feasible.

We agree with the reviewer’s comment and have included APOP analysis on the TcoPYK T state crystal structure (see also reply to Comment 3.1). Briefly, APOP works by detecting pockets and then perturbing each pocket in the protein's elastic network (GNM) by adding stiffer springs between the surrounding residues. The pockets are scored and ranked based on the calculated shifts in the eigenvalues of the global GNM modes and their local hydrophobic densities, thereby also considering the pocket’s surface accessibility, which renders it suitable for the identification of allosteric (and druggable) pockets. The APOP analysis identifies pockets overlapping with the sdAb42 epitope as highly ranking allosteric ligand binding pockets. The data have been summarized in an additional supplementary figure (Figure 4 – figure supplement 1). The manuscript also contains details on the performed APOP analysis in the Materials and Methods section.

Comment 2.4: The authors comment on the modest growth inhibition, and refer to the need to achieve over 88% reduction in Vmax of PYK to see a strong effect, something that may or may not be achieved in the cell-based model (no target-engagement or functional readout provided). The slow binding model and switch of species are also raised as potential explanations. While these may be plausible explanations, they are not tested which leaves us with limited evidence to support targeting the allosteric site on PYK.

In our understanding of this remark, we believe it be related to Comments 2.1 and 2.2 and thus refer to our answers formulated above.

Comment 2.5: The evidence to support an allosteric mechanism is derived from structural studies, including the in silico allosteric network predictions. Unfortunately, standard enzyme kinetics mode of inhibition studies are missing. Such studies could distinguish uncompetitive from non-competitive behaviour and strengthen the claim that sdAb42 locks the enzyme complex in the apo form.

We agree with the referee that a thorough kinetic analysis could distinguish between uncompetitive (i.e., sdAb only binds to the enzyme if substrate is bound) or non-competitive (i.e., sdAb can bind to apo enzyme and substrate-bound enzyme) inhibition. In both cases, however, this would correspond to an allosteric mechanism of inhibition. Although such a thorough kinetic analysis would be interesting in its own right, we would like to argue that this type of very detailed kinetics is outside the scope of this paper. This is especially the case taking into account that this analysis could be complicated by the slow-onset inhibition behavior.

Comment 2.6: As general comment, the graphical representation of the data could be improved in line with recent recommendations: https://journals.plos.org/plosbiology/article?id=10.1371/journal.pbio.1002128, https://elifesciences.org/inside-elife/5114d8e9/webinar-report-transforming-data-visualisation-to-improve-transparency-and-reproducibility.Bar-charts for potency are ideally presented as dot plots, showing the individual data points, or box plots with datapoints shown.Images in Figure 7 show significant heterogeneity of nanobody expression, but the extent of this can not be gleaned from Figure 7B. It would be much better to use box plots or violin plots for each cell line on this figure panel. The same applies to Figure 10.

We thank the reviewer for these suggestions but have taken the decision not to act upon these as the other reviewers explicitly mentioned that our figures are very clear.

**Recommendations for authors:**
Please find below some minor comments:Comment 2.7: Line 24: "increasing number of drug failures": This does not really reflect the current situation for human African trypanosomiasis, with NECT treatment retaining efficacy, fexinidazole now being registered, and acoziborole progressing towards registration. It may be worth considering focusing the introduction more on Nagana, as all Trypanosome species used in the paper are animal infective, and the nanobody was discovered for T. congolense.

We refer to our answer formulated in response to Comment 1.1.

Comment 2.8: Line 55: "alarming number of reports describing ..." While resistance is a big problem, this mainly applies to malaria, bacterial and fungal diseases. For kinetoplastids, the number of reports describing resistance in the clinic is pretty limited. However, the drug discovery pipeline for these diseases is sparse, so I definitely agree there is a need to develop new compounds with differentiated mechanisms.

We agree with the reviewer and have slightly adapted our wording here as follows.

“Unfortunately, a number of reports describe treatment failure or parasite resistance to the currently available drugs (De Rycker et al., 2018).”

Comment 2.9: This manuscript is about pyruvate kinase, but the enzyme is not properly introduced. I suggest a short paragraph introducing PYK at line 77 (without duplicating Figure 1), covering its role in glycolysis, the importance of pyruvate, any essentiality data from the literature, and any known inhibitors.

We refer to our answer formulated in response to Comment 1.3.

Comment 2.10: Figure 6: For the top insets it would be useful to somehow show the increasing antibody concentration, either by using a changing intensity or size for each line.

We thank the reviewer for this suggestions, but decided not to act upon it as we found that the inclusion of this information in the figure made it “too crowded”, which is why we opted to provide this information in the figure legend.

“Only a subset of the traces is shown for the sake of clarity. The following curves are shown (from bottom to top): TcoPYK (0.15 nM sdAb42, 500 nM sdAb42, 750 nM sdAb42, 1000 nM sdAb42, 1500 nM sdAb42, 2000 nM sdAb42, no enzyme control), LmePYK (5 nM sdAb42, 750 nM sdAb42, 1250 nM sdAb42, 1500 nM sdAb42, 2500 nM sdAb42, 3000 nM sdAb42, no enzyme control), and TbrPYK (1 nM sdAb42, 1000 nM sdAb42, 1750 nM sdAb42, 2000 nM sdAb42, 3500 nM sdAb42, 4000 nM sdAb42, no enzyme control).”

Comment 2.11: You refer to the curves as biphasic, but they look like 1st order kinetics, and there are no clear 1st and 2nd phases (or at least they are not marked). It may be more appropriate to label these as non-linear.

We agree that the term “biphasic” is potentially an over-simplification of the actual situation. What we mean is that the formation of product as a function of time ([P] versus [t] curve) is not linear at short time ranges but evolves from an initial “weakly inhibited” rate (v_0_) to a “strongly inhibited” steady-state rate (v_ss_). This conversion from v_0_ to v_ss_ indeed occurs in a fashion following single exponential behavior. With the term “biphasic” we thus meant a non-linear phase (before v_ss_ is reached) followed by a linear phase (after v_ss_ is reached). To avoid any confusion, we replaced the term “biphasic” by “non-linear”.

Comment 2.12: IC50s - would be useful to provide a comparison with IC50s generated in the pre-incubation experiments - is the antibody less potent without pre-incubation? I could not find IC50s for the pre-incubation experiments shown in Figure 2.

We determined an IC50 value for sdAb42 against TcoPYK under pre-incubation conditions, but initially decided not to include this into the manuscript. We agree with the reviewer that a comparison between IC50 values determined under pre- and post-incubation conditions would be of interest, and have therefore included the pre-incubation IC50 data for TcoPYK in Figure 2 (panel B). The data indeed show that sdAb42 far more efficiently inhibits an enzyme that is not continuously cycling between R and T states (IC50 values of 15 nM and 359 nM under pre- and post-incubation conditions, respectively). This is now discussed in the results section of the manuscript. We did not determine IC50 values for sdAb42 against TbrPYK and LmePYK under pre-incubation conditions, but suspect that a similar observation will be made upon comparing these values to IC50 under post-incubation conditions.

**REVIEWER 3:**
Summary:Out of the 20 Neglected Tropical Diseases (NTD) highlighted by the WHO, three are caused by members of the trypanosomatids, namely Leishmanaisis, Trypanosomiasis, and Chagas disease. Trypanosomal glycolytic enzymes including pyruvate kinase (PyK) have long been recognised as potential targets. In this important study, single-chain camelid antibodies have been developed as novel and potent inhibitors of PyK from the T, congolense. To gain structural insight into the mode of action, binding was further characterised by biophysical and structural methods, including crystal structure determination of the enzyme-nanobody complex. The results revealed a novel allosteric mechanism/pathway with significant potential for the future development of novel drugs targeting allosteric and/or cryptic binding sites.Strengths:This paper covers an important area of science towards the development of novel therapies for three of the Neglected Tropical Diseases. The manuscript is very clearly written with excellent graphics making it accessible to a wide readership beyond experts. Particular strengths are the wide range of experimental and computational techniques applied to an important biological problem. The use of nanobodies in all areas from biophysical binding experiments and X-ray crystallography to in-vivo studies is particularly impressive. This is likely to inspire researchers from many areas to consider the use of nanobodies in their fields.Weaknesses:There is no particular weakness, but I think the computational analysis of allostery, which basically relies on a single server could have been more detailed.
**Recommendations for authors:**
Overall an excellent paper, there are just a couple of points that the authors could consider, if time allows.Comment 3.1: As mentioned above the computational analysis of allostery appears to be based on a single server based on coordinates alone with no in-depth analysis. It would be extremely interesting to see if more sophisticated methods based on elastic network model and/or molecular dynamics simulation gave similar results. I realize that this would require quite a lot of work though.

We agree with the reviewer’s comment and have complemented the perturbation analysis (previously presented in the manuscript) with dGNM and APOP analyses to identify allosteric communication pathways and allosteric binding pockets, respectively. dGNM, which is based on transfer entropy, allowing for a detailed characterization of the dynamic coupling and information transfer between residues. Meanwhile, APOP employs a perturbation-based approach to detect and rank allosteric pockets. The findings are in good agreement with the previously presented perturbation data and have been summarized in an additional supplementary figure (Figure 4 – figure supplement 1). The manuscript also contains details on the performed transfer entropy and APOP analyses in the Materials and Methods section.

Comment 3.2: The figures are excellent and really help the reader - with the exception of the screenshots (Figure 8). Using pymol or chimera (or any other more expensive commercial package) would really help the reader and will not take much time.

We agree with the referee that this is not the most beautiful figure. However, we find the quality and clarity of the figure to be adequate for its purpose (i.e., a supplemental figure).

Comment 3.3: Finally, I would have liked to see at least the PDB validation files. This is a highly regarded and experienced team, nevertheless, the resolution is rather mediocre. As the crystal coordinates were used as input for the computational, any experimental inaccuracies will affect the computational results.

We agree with the reviewer that we could have provided the validation report together with the submitted manuscript and we apologise for the inconvenience. The validation reports will be released together with the structures following final manuscript publication. Regarding the resolution of the crystal structures, we agree with the reviewer’s comment, but we obviously employed data sets from our best diffracting crystals and could not obtain a higher resolution despite our best efforts.